biomedical engineering/biomechanics/
mathematical modelling

myofibre architecture, bi-ventricular model,
rule-based approach, structural tensor,
fibre dispersion

**Author for correspondence:**
Hao Gao
e-mail: hao.gao@glasgow.ac.uk

# Effect of myofibre architecture on ventricular pump function by using a neonatal porcine heart model: from DT-MRI to rule-based methods

## Debao Guan[1], Jiang Yao[2], Xiaoyu Luo[1] and Hao Gao[1]

[1]School of Mathematics & Statistics, University of Glasgow, Glasgow, UK
[2]Dassault Systemes, Johnston, RI, USA

XL, 0000-0002-8753-4210; HG, 0000-0001-6852-9435

Myofibre architecture is one of the essential components when constructing personalized cardiac models. In this study, we develop a neonatal porcine bi-ventricle model with three different myofibre architectures for the left ventricle (LV). The most realistic one is derived from *ex vivo* diffusion tensor magnetic resonance imaging, and other two simplifications are based on rule-based methods (RBM): one is regionally dependent by dividing the LV into 17 segments, each with different myofibre angles, and the other is more simplified by assigning a set of myofibre angles across the whole ventricle. Results from different myofibre architectures are compared in terms of cardiac pump function. We show that the model with the most realistic myofibre architecture can produce larger cardiac output, higher ejection fraction and larger apical twist compared with those of the rule-based models under the same pre/after-loads. Our results also reveal that when the cross-fibre contraction is included, the active stress seems to play a dual role: its sheet-normal component enhances the ventricular contraction while its sheet component does the opposite. We further show that by including non-symmetric fibre dispersion using a general structural tensor, even the most simplified rule-based myofibre model can achieve similar pump function as the most realistic one, and cross-fibre contraction components can be determined from this non-symmetric dispersion approach. Thus, our study highlights the importance of including myofibre dispersion in cardiac modelling if RBM are used, especially in personalized models.

# 1. Introduction

Cardiac disease remains the leading cause of mortality and morbidity worldwide, as a result, extensive research has been carried out to develop computational cardiac models to understand mechanical behaviours of the heart [1–3]. For instance, finite-element (FE) method has been widely used to model heart function physiologically or pathologically, and to develop novel therapies [2,4,5]. The remaining challenges are to deal with the complex geometry, myofibre structure and material characterization of the myocardium [6,7]. Recent reviews on heart modelling can be found in [3,8,9].

The spatial architecture of myofibres plays a central role in electrical propagation, myocardial expansion and contraction [10]. Early studies relied on fibre dissections and histological slices [11] to determine local fibre structure. Currently, the cardiac fibres can be imaged via diffusion tensor magnetic resonance imaging (DT-MRI) [12] that allows a direct description of the three-dimensional myofibre architecture. To reconstruct myofibres in computational models, two different approaches have been developed. One is directly mapping myofibres from *ex/in vivo* datasets to the models, i.e. reconstructing models directly from DT-MRI [6], or using atlas-based methods to warp DT-MRI data into different models [13]. The other approach is the rule-based method (RBM), in which myofibres rotate from endocardium to epicardium with prescribed angles concerning the circumferential direction, varied linearly across the wall in most of the studies [14–16]. One key step in RBM is to parametrize the normalized wall thickness ($\bar{e}$) in order to assign local fibre angles, from $\bar{e} = 0$ at endocardial surface to $\bar{e} = 1$ at epicardial surface. With measured fibre angles at endocardium $\theta^{\mathrm{endo}}$ and epicardium $\theta^{\mathrm{epi}}$, the local fibre angle can then be assigned by varying linearly or nonlinearly with $\bar{e}$. Bayer *et al.* [17] proposed a Laplace–Dirichlet RBM, in which the circumferential-radial (transmural)-longitudinal directions and normalized wall thickness are determined by solving a series of Laplace equations. They demonstrated that the Laplace-Dirichlet rule-based fibre could achieve almost identical electrical activation patterns in a whole heart model as a DT-MRI-based model. Additionally, regionally varied RBM has been developed to take into account spatial variations of myofibre rotation angles [18].

Three-dimensional FE mechanics models of the heart have been used extensively to investigate the role of myofibre architecture in cardiac function under normal and abnormal function, including ischaemia, ventricular pacing, myofibre disarray and heart failure. For example, By using a rule-based approach for myofibre reconstruction in an left ventricle (LV) model, Wang *et al.* [15] found that changes in myofibre rotation angle can dramatically affect the stress and strain distributions during diastole. Using a bi-ventricular model, Palit *et al.* [19] also demonstrated that changes in myofibre angle can significantly affect myofibre stress–strain distribution within the LV wall in diastole. Pluijmert *et al.* [20] found that a change of 8° in myofibre orientation along transmural direction can cause a considerable increase in cardiac pump work (17%). In a recent study, Gil *et al.* [21] compared three different myofibre architectures in an electromechanics bi-ventricular model, one is from a DT-MRI dataset [22], the other two are reconstructed using a rule-based approach [14] with histologically measured myofibre angles [23]. Their results showed that the model with realistic myofibre structure from DT-MRI produces functional scores much closer to healthy ranges than rule-based approaches. By using the polynomial chaos expansion method, Rodríguez-Cantano *et al.* [24] studied the uncertainty in myofibre orientation and demonstrated that a realistic myofibre structure is necessary for a personalized cardiac model, such as DT-MRI-acquired myofibres.

Furthermore, myofibrils do not align perfectly along one direction at any location within a ventricular wall, but dispersed as reported by Ahmad *et al.* [25], who measured in-plane and out-of-plane myofibre and collagen fibre dispersion using two-photon-excited fluorescence and second harmonic generation microscopy on neonatal heart samples. To incorporate fibre dispersion in material constitutive law, Gasser *et al.* [26] introduced a structural tensor to account for collagen fibre dispersion in arterial tissue, they assumed a rotational symmetry for fibre distribution and a compact form of characterizing fibre dispersion was then given with one dispersion parameter, the so-called $\kappa$-model. In a series of studies [27,28], Holzapfel and co-workers used this generalized structural tensor to characterize the passive response of fibre-reinforced soft tissues. Later on, Pandolfi and co-workers [29–31] extended Gasser's general structural tensor approach by including the second-order term of the Taylor expansion on the mean invariant along the fibre direction, to improve the accuracy of a structural tensor with large dispersions. In a recent study, Melnik [32] further extended the generalized structural tensor to include fibre dispersion in a coupled strain invariant.

Although there are several studies on passive constitutive responses of soft tissue [26,27], very few studies included fibre dispersion in active contraction models for the myocardium. There are two commonly used approaches for modelling active contraction in biological tissue: the active stress formulation [5,6,18,33] and the active strain formulation [9,34,35]. In the active stress formulation, the total stress tensor is decomposed into passive and active parts [36]. This approach has been widely used in personalized

cardiac modelling because of its easy implementation, and the fact that there are abundant experimental data for the parameter calibration [5,6,18,33]. In the active strain approach, the total deformation gradient $\mathbf{F}$ is multiplicatively decomposed into an elastic part ($\mathbf{F}^{\mathrm{pass}}$) for passive response and an activation part ($\mathbf{F}^{\mathrm{act}}$), which could be more inherent to the 'sliding filament theory' [34]. The same structural tensor for the passive response could be linked to $\mathbf{F}^{\mathrm{act}}$ to account for the active response [35]. This seems to be an elegant approach, though fitting personalized parameters to experimental data remains a challenge [34]. It is for this reason, that the active stress approach is still adopted here. To take into account active contraction caused by dispersed myofibres, Guccione and co-workers introduced cross-fibre active contraction in cardiac models [33,37] based on experiments by Lin & Yin [38]. Recently, Sack *et al.* [6] inversely determined cross-fibre contraction ratio in a healthy porcine heart and a failing heart. It has been argued that cross-fibre active contraction may be related to myofibre dispersion. However, no detailed studies have reported this connection. Eriksson *et al.* [18] incorporated myofibre dispersion in both the passive and active mechanics in an electromechanically coupled idealized left ventricular model. Their model, based on the $\kappa$-model [26], showed that large dispersion in the diseased heart could greatly affect the ventricular pump function. On the other hand, Ahmad's study [25] demonstrated that in-plane dispersion is different from out-of-plane dispersion, which suggests the rotational symmetry assumption used in the $\kappa$-model may not be appropriate. Therefore, for the active stress formulation, a better approach would be to use the non-symmetric dispersion model developed in [27]. In this study, myocardial contraction is modelled following the active stress approach similar to that in [6].

Overall, there is a lack of studies on how different myofibre generation approaches, DT-MRI derived or RBM, affecting ventricular pump functions. One particular question is whether the difference between DT-MRI- and RBM-based models can be rectified using a proper consideration of fibre dispersion. We hypothesize that incorporating a non-symmetrical dispersed active tension model in an RBM-generated myofibre architecture can approximate the DT-MRI-based approach when simulating the heart pump function.

# 2. Material and methods

## 2.1. Geometry and fibre construction

A three-dimensional FE bi-ventricular model from [39] is used in this study (figure 1*a*), which is reconstructed from a computed tomography (CT) data of a neonatal porcine heart. Details of the data acquisition can be found in [40]. The three-dimensional CT data are first segmented using Seg3D,[1] then the boundary contours are exported into SolidWorks (Dassault Systemes, MA, USA) for geometry reconstruction, and meshed (figure 1*a*) using ICEM (ANSYS, Inc., PA, USA).

Because the myofibre structure of the neonatal porcine heart is not available, it is interpolated from a canine heart obtained from the public dataset of Cardiovascular Research Grid[2] [22]. We first reconstruct a bi-ventricular geometry for the canine heart with myofibres extracted from the primary eigenvector of the DT-MRI tensors, as shown in figure 1*b*. Clearly, the neonatal bi-ventricle geometry is different from the canine geometry, as shown in figure 1. Therefore, we cannot directly interpolate the measured canine myofibre structure for the neonatal bi-ventricle model. Instead, *Deformetrica*[3] is employed to register the two bi-ventricular geometries by warping a template ($C_\alpha$: the canine bi-ventricle) to a target ($C_\beta$: the neonatal porcine heart) by minimizing a loss function that measures the distance between the template and target. *Deformetrica* is an open-source package based on a large deformation diffeomorphic metric mapping (LDDMM) framework [41,42]; further details about *Deformetrica* are given in the electronic supplementary material.

After warping $C_\alpha$ into $C_\beta$, the displacement fields $\mathbf{u}$ for all nodes on the external surface of $C_\alpha$ are obtained, denoting $\mathbf{u}_{\mathrm{Ex}}^{\mathrm{LDDMM}}$ as shown in figure 1*c*. The displacement vectors on the nodes lying within the ventricular wall are then interpolated by solving a Laplace system with Dirichlet boundary conditions (equation (2.1)) in *Fenics*,[4]

$$\begin{cases} \nabla^2 \mathbf{u} = 0, \\ \mathbf{u} = \mathbf{u}_{\mathrm{Ex}}^{\mathrm{LDDMM}} \quad \text{at external surface.} \end{cases} \tag{2.1}$$

---

[1] See http://www.sci.utah.edu/cibc-software/seg3d.html.

[2] See http://cvrgrid.org/data/ex-vivo.

[3] See http://www.deformetrica.org/.

[4] See https://fenicsproject.org/.

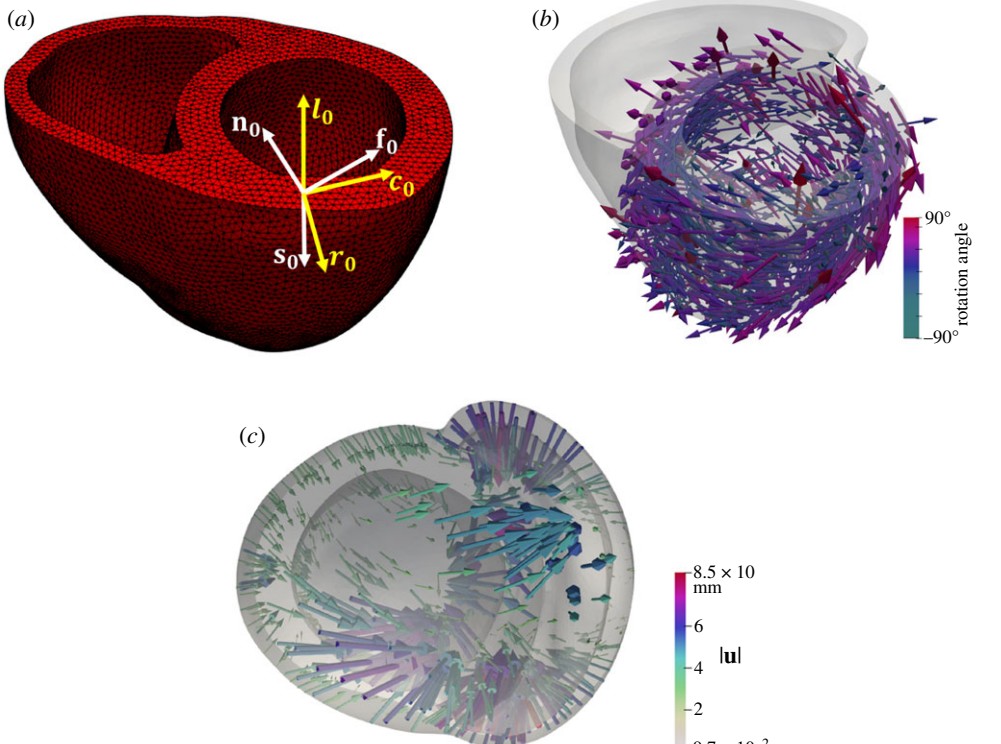

**Figure 1.** (a) The reconstructed bi-ventricle neonatal heart geometry from three-dimensional CT data (263 972 linear tetrahedral elements and 50 640 nodes). Local coordinate system, $\mathbf{f}_0$, $\mathbf{s}_0$, $\mathbf{n}_0$ are the conventional fibre–sheet–normal system, in which $\mathbf{f}_0$ is the mean fibre direction, $\mathbf{s}_0$ is the sheet direction usually formed by 4–6 myocytes, and in general along the transmural direction from endocardium to epicardium, and $\mathbf{n}_0$ is the sheet-normal direction. $\mathbf{c}_0$, $\mathbf{r}_0$, $\mathbf{l}_0$ are the local circumferential-radial-longitudinal system. (b) The reconstructed canine heart (252 713 linear tetrahedral elements and 49 460 nodes) with corresponding DT-MRI fibres. (c) Displacement vectors ($\mathbf{u}$) for warping the canine geometry to the porcine heart, coloured by the magnitude of $\mathbf{u}$.

Following the finite deformation theory, the deformation gradient of warping the canine bi-ventricle model into the porcine model is

$$\mathbf{F} = \nabla\mathbf{u} + \mathbf{I}, \tag{2.2}$$

in which $\mathbf{I}$ is the identity matrix. Note that $\mathbf{F}$ and $\mathbf{u}$ are associated with the canine bi-ventricle model. Myofibre orientation in the warped canine model is

$$\mathbf{f}_{\text{warp}}^{\text{canine}} = \frac{\mathbf{F}\,\mathbf{f}_{\text{template}}^{\text{canine}}}{|\mathbf{F}\,\mathbf{f}_{\text{template}}^{\text{canine}}|} \tag{2.3}$$

where $\mathbf{f}_{\text{template}}^{\text{canine}}$ is the unit myofibre direction from the DT-MRI canine dataset. Finally myofibres in the porcine model $\mathbf{f}_0$ are assigned according to the nearest neighbours between the warped canine and porcine geometries, such that

$$\mathbf{f}_0 = \mathbf{f}^{\text{porcine}}(\mathbf{x}^{\text{porcine}}) \approx \delta(\mathbf{x}^{\text{porcine}} - \mathbf{x}_{\text{warp}}^{\text{canine}})\,\mathbf{f}_{\text{warp}}^{\text{canine}}(\mathbf{x}_{\text{warp}}^{\text{canine}}), \tag{2.4}$$

in which $\mathbf{x}^{\text{procine}}$ is a position vector in the porcine model, and $\mathbf{x}_{\text{warp}}^{\text{canine}}$ is the position vector in the warped canine model. The sheet direction $\mathbf{s}_0$ is defined transmurally across the wall, and the sheet-normal is $\mathbf{n}_0 = \mathbf{f}_0 \times \mathbf{s}_0$.

We further generate two different myofibre structures in the left side of the bi-ventricle using a rule-based approach [15], septum included. By projecting $\mathbf{f}_0$ into the $\mathbf{c}_0 - \mathbf{l}_0$ plane to have $\mathbf{f}_0^{\parallel}$, we define the myofibre angle as the angle between $\mathbf{f}_0^{\parallel}$ and $\mathbf{c}_0$, as shown in figure 2a. The average myofibre angles in the porcine model are then summarized at endocardium ($\theta_{\text{endo}}^{\text{ave}}$) and epicardium ($\theta_{\text{epi}}^{\text{ave}}$) in two ways: (i) across the whole LV, and (ii) at each ventricular segment according to the AHA17 (American Heart Association) definition [43] as shown in figure 2b,c based on right ventricular insertion points. A rule-based approach is used to generate two different myofibre structures: (i) one set of myofibre rotation angles varies linearly from

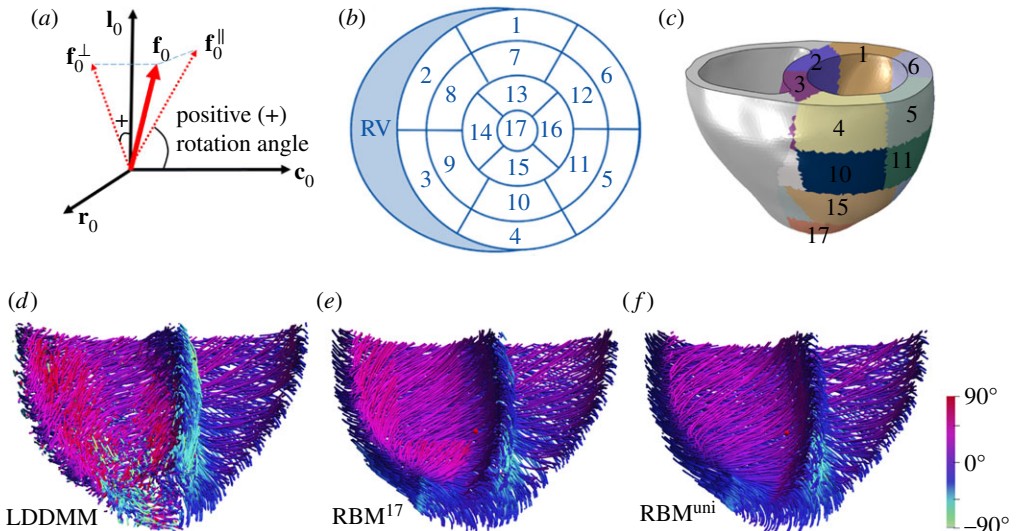

**Figure 2.** Myofibre rotation angle definition ($a$), which is the angle between $\mathbf{f}_0^{\|}$ and $\mathbf{c}_0$. $\mathbf{f}_0^{\|}$ (in-plane) and $\mathbf{f}_0^{\perp}$ (out-of-plane) are the projections of $\mathbf{f}_0$ in $\mathbf{c}_0 - \mathbf{l}_0$ and $\mathbf{l}_0 - \mathbf{r}_0$ planes, respectively, ($b$) AHA 17 segments definition in a bullseye view and ($c$) in the porcine model. Three different myofibre architectures are generated, they are ($d$) LDDMM derived, ($e$) RBM[17] and ($f$) RBM$^{\text{uni}}$.

**Table 1.** Average myofibre rotation angles (°) at endocardium and epicardium according to the AHA17 definition, and the set of angles for the RV.

| section | 1 | 2 | 3 | 4 | 5 | 6 | 7 | 8 | 9 |
|---|---|---|---|---|---|---|---|---|---|
| endocardium | 20 | 40 | 30 | 40 | 60 | 40 | 40 | 60 | 30 |
| epicardium | −20 | −40 | −40 | 0 | −20 | −20 | −40 | −40 | −30 |
| section | 10 | 11 | 12 | 13 | 14 | 15 | 16 | 17 | RV |
| endocardium | 40 | 60 | 40 | 60 | 30 | 80 | 60 | 10 | 40 |
| epicardium | −20 | −20 | −40 | −40 | −30 | −20 | −40 | −10 | −30 |

endocardium to epicardium for the whole LV; (ii) for each AHA17 segment, myofibre rotates linearly based on the average rotation angles from that segment, which means myofibre angles are different at different segments. Note that the myofibre structure in the right ventricle (RV) of the bi-ventricle model, excluding the septum, is generated by the same rule-based approach but using one set of rotation angles due to the lack of DT-MRI data for the right side. We further assume the myofibre rotation angles at RV are the same as the angles when averaged across the whole LV.

With these three myofibre structures generated (figure 2$d$–$f$), we consider the following cases:

— case LDDMM: the LV with the mapped *ex vivo* DT-MRI acquired myofibre architecture (equation (2.4));
— case RBM[17]: myofibre rotates linearly from endocardium to epicardium for each LV segment according to the average rotation angles at 17 segments, derived from case LDDMM; table 1 lists the myofibre rotation angles at each segment, including the angles for the RV;
— case RBM$^{\text{uni}}$: myofibre uniformly rotates between endocardium and epicardium in the whole LV with one set of the average rotation angles (endocardium: 40°, epicardium: −30°), which are also derived from case LDDMM.

Note that case RBM[17] has a heterogeneous myofibre structure in the whole LV but homogeneous within each segment. We also have not smoothed rotation angles between segments since those variations are within the range of local angle variations in case LDDMM as suggested in figure 4$a$. Thus, case RBM[17] is a simplification of case LDDMM. Case RBM$^{\text{uni}}$ has the same myofibre structure across the whole LV, a further simplification compared with case RBM[17].

## 2.2. Constitutive model

### 2.2.1. Passive stress response

The passive behaviour of myocardium is described by a strain-invariant-based function [39], which is reduced from the model proposed by Holzapfel & Ogden [7] by fitting to an experimental study of neonatal porcine myocardium [40]. The strain energy function consists of a deviatoric ($\Psi_{\text{dev}}$) and a volumetric ($\Psi_{\text{vol}}$) parts,

$$
\begin{aligned}
\Psi_{\text{dev}} = {} & \frac{a}{2b}\exp\left[b(\bar{I}_1 - 3)\right] \\
& + \sum_{i=\text{f,n}}\frac{a_i}{2b_i}\{\exp\left[b_i(\max(\bar{I}_{4i}, 1) - 1)^2\right] - 1\} \\
& + \sum_{ij=\text{fs,fn}}\frac{a_{ij}}{2b_{ij}}\left[\exp\left(b_{ij}\bar{I}_{8ij}^2\right) - 1\right]
\end{aligned}
\tag{2.5}
$$

and
$$
\Psi_{\text{vol}} = \frac{1}{D}\left(\frac{J^2 - 1}{2} - \ln(J)\right),
$$

where $a, b, a_i, b_i, a_{ij}, b_{ij}$ are material constants and $D$ is a multiple of the bulk modulus $K$, i.e. $D = 2/K$. $J = \det(\mathbf{F})$, $\mathbf{F} = J^{1/3}\bar{\mathbf{F}}$ and $\bar{\mathbf{C}} = \bar{\mathbf{F}}^{\mathrm{T}}\bar{\mathbf{F}}$. The isochoric invariants are defined as $\bar{I}_1 = \text{trace}(\bar{\mathbf{C}})$, $\bar{I}_{4\text{f}} = \mathbf{f}_0 \cdot \bar{\mathbf{C}}\mathbf{f}_0$, $\bar{I}_{4\text{n}} = \mathbf{n}_0 \cdot \bar{\mathbf{C}}\mathbf{n}_0$, $\bar{I}_{8\text{fs}} = \mathbf{f}_0 \cdot \bar{\mathbf{C}}\mathbf{s}_0$ and $\bar{I}_{8\text{fn}} = \mathbf{f}_0 \cdot \bar{\mathbf{C}}\mathbf{n}_0$, in which $\mathbf{f}_0, \mathbf{s}_0, \mathbf{n}_0$ are the myofibre, sheet and sheet-normal directions in the reference state. In this study, we assume the collagen fibres follow the layered myocyte structure. Thus, myofibres represent both myocyte and collagen fibres. The max () in equation (2.5) will ensure the collagen fibres can only bear load when in tension. The passive Cauchy stress tensor is given by

$$
\begin{aligned}
\boldsymbol{\sigma}^{\mathrm{P}} = {} & p_{\text{vol}}\mathbf{I} + 2J^{-1}[\bar{\psi}_1\,\text{dev}\,\bar{\mathbf{b}} + \bar{\psi}_{4\text{f}}\,\text{dev}\,(\bar{\mathbf{f}} \otimes \bar{\mathbf{f}}) + \bar{\psi}_{4\text{n}}\,\text{dev}\,(\bar{\mathbf{n}} \otimes \bar{\mathbf{n}}) \\
& + \frac{1}{2}\bar{\psi}_{8\text{fs}}\,\text{dev}\,(\bar{\mathbf{f}} \otimes \bar{\mathbf{s}} + \bar{\mathbf{s}} \otimes \bar{\mathbf{f}}) + \frac{1}{2}\bar{\psi}_{8\text{fn}}\,\text{dev}\,(\bar{\mathbf{f}} \otimes \bar{\mathbf{n}} + \bar{\mathbf{n}} \otimes \bar{\mathbf{f}})],
\end{aligned}
\tag{2.6}
$$

in which $\bar{\psi}_i = \partial\Psi_{\text{dev}}/\partial\bar{I}_i$, $i \in \{1, 4\text{f}, 4\text{n}, 8\text{fs}, 8\text{fn}\}$, $\bar{\mathbf{f}} = \bar{\mathbf{F}}\mathbf{f}_0$, $\bar{\mathbf{s}} = \bar{\mathbf{F}}\mathbf{s}_0$, $\bar{\mathbf{n}} = \bar{\mathbf{F}}\mathbf{n}_0$, $\bar{\mathbf{b}} = \bar{\mathbf{F}}\bar{\mathbf{F}}^{\mathrm{T}}$, $p_{\text{vol}} = \partial\Psi_{\text{vol}}/\partial J$, and $\text{dev}(\bullet) = (\bullet) - (1/3)[(\bullet)\text{:}\mathbf{I}]\,\mathbf{I}$ denotes the deviatoric operator.

### 2.2.2. Active stress

Biaxial investigations on actively contracting rabbit myocardium [38] suggest that a large portion of active stress exists in cross-fibre direction. This has motivated computational efforts to include a proportion of the active stress to the cross-fibre direction when RBM generated myofibres are used [4,37]. In this study, we employ the active stress approach for myocardial active stress along the myofibre, sheet and sheet-normal directions

$$
\boldsymbol{\sigma}^{\mathrm{a}} = n_{\text{f}}\,T_{\text{a}}\,\hat{\mathbf{f}} \otimes \hat{\mathbf{f}} + n_{\text{s}}\,T_{\text{a}}\,\hat{\mathbf{s}} \otimes \hat{\mathbf{s}} + n_{\text{n}}\,T_{\text{a}}\,\hat{\mathbf{n}} \otimes \hat{\mathbf{n}},
\tag{2.7}
$$

in which $\hat{\mathbf{f}} = \mathbf{f}/|\mathbf{f}|$, $\hat{\mathbf{s}} = \mathbf{s}/|\mathbf{s}|$ and $\hat{\mathbf{n}} = \mathbf{n}/|\mathbf{n}|$, $n_{\text{f}}$, $n_{\text{s}}$ and $n_{\text{n}}$ (all positive and sum up to 1) are the proportions of the active tension in their respective directions. $T_{\text{a}}$ is the active tension generated along the myofibre direction, which is described by a time-varying elastance model that has been described extensively in the literature [6,36,37]

$$
T_{\text{a}}(t, l) = \frac{T_{\max}}{2}\frac{\text{Ca}_0^2}{\text{Ca}_0^2 + \text{ECa}_{50}^2(l)}(1 - \cos(\omega(t, l))),
\tag{2.8}
$$

where $T_{\max}$ is the maximum allowable active tension, $\text{Ca}_0$ is the peak intracellular calcium concentration, $\text{ECa}_{50}$ represents length-dependent calcium sensitivity, $t$ is time and $l$ is myofibre stretch. Further details are provided in the electronic supplementary material.

We assume the cross-fibre contraction in the RV is zero, i.e. $n_{\text{f}} = 1$, $n_{\text{s}} = 0$, and $n_{\text{n}} = 0$. This is because RV has a much thinner wall thickness, and Ahmad et al. [25] reported the fibre dispersion in the RV is much less than in the LV (9.3° versus 19.2°). We also performed simulations for the RBM$^{\text{uni}}$ case, using the LV's non-zero cross-fibre contraction for the RV. Our results show the differences of ejection fraction are 0.7% and 4.1% for the LV and RV, respectively. Thus assuming no cross-fibre contraction for the RV seems to be reasonable.

As for the LV (septum is included), since DT-MRI-derived myofibres are naturally dispersed in case LDDMM (figure 2), we set $n_{\text{f}} = 1$, $n_{\text{s}} = 0$ and $n_{\text{n}} = 0$. But for the RBM cases, it is necessary to include cross-

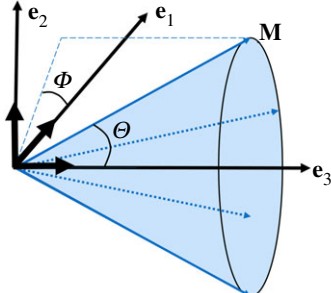

**Figure 3.** A unit vector $\mathbf{M}(\Theta, \Phi)$ representing a fibre direction defined by $\Theta$ and $\Phi$ with respect to a Cartesian system $\mathbf{e}_1$, $\mathbf{e}_2$ and $\mathbf{e}_3$. The plane spanned by $\mathbf{e}_2$–$\mathbf{e}_3$ is in-plane while out-of-plane is $\mathbf{e}_1$–$\mathbf{e}_2$. The mean myofibre direction is along $\mathbf{e}_3$.

fibre active tension, and $n_s$ and $n_n$ are calculated based on a dispersed fibre structure tensor. This will be explained in the following section.

### 2.2.3. Determination of $n_f$, $n_n$ and $n_s$ using DT-MRI derived myofibres for case RBM$^{uni}$

We first introduce $\{\mathbf{e}_1, \mathbf{e}_2, \mathbf{e}_3\}$ to denote the axes of a Cartesian coordinate system as shown in figure 3, and then define myofibre direction of the reference configuration to be $\mathbf{M}$ with a density distribution $\rho(\mathbf{M})$. $\mathbf{M}$ can be further characterized by two angles $\Theta \in [0, \pi]$ and $\Phi \in [0, 2\pi]$ (figure 3), that is

$$\mathbf{M}(\Theta, \Phi) = \sin\Theta\cos\Phi\,\mathbf{e}_1 + \sin\Theta\sin\Phi\,\mathbf{e}_2 + \cos\Theta\,\mathbf{e}_3. \tag{2.9}$$

$\Theta$ is the angle between $\mathbf{e}_3$ and $\mathbf{M}$, and $\Phi$ is the angle between $\mathbf{e}_1$ and the projected vector of $\mathbf{M}$ in the $\mathbf{e}_1$–$\mathbf{e}_2$ plane. We assume the dispersions in different planes are essentially independent [44], i.e.

$$\rho(\mathbf{M}) = \rho(\Theta, \Phi) = \rho_{op}(\Phi)\,\rho_{in}(\Theta), \tag{2.10}$$

in which $\rho_{op}(\Phi)$ describes the out-of-plane dispersion, and $\rho_{in}(\Theta)$ describes the in-plane dispersion. Note in the ventricular model, in-plane is the plane defined by $\mathbf{c}_0 - \mathbf{l}_0$, and out-of-plane is the plane defined by $\mathbf{l}_0 - \mathbf{r}_0$. This is consistent with experimental studies when measuring in-/out-of-plane fibre angles [25,45]. The normalization of $\rho(\Theta, \Phi)$ over a unit sphere requires

$$\frac{1}{N}\int_0^{2\pi}\int_0^{\pi}\rho_{op}(\Phi)\,\rho_{in}(\Theta)\sin\Theta\,d\Theta\,d\Phi = 1, \tag{2.11}$$

in which $N$ is a normalization factor.

When there is no dispersion, the structure tensor $\mathbf{M}\otimes\mathbf{M}$ can be directly used for constructing $I_{4f} = \mathbf{C}:\mathbf{M}\otimes\mathbf{M}$ and active stress tensor $T_a\,\mathbf{M}\otimes\mathbf{M}/I_{4f}$. With dispersion, a generalized structure tensor $\mathbf{H}$ can be defined over an unit sphere [18,26,27],

$$\mathbf{H} = \frac{1}{N}\int_0^{2\pi}\int_0^{\pi}\rho_{op}(\Phi)\,\rho_{in}(\Theta)\sin\Theta\,\mathbf{M}\otimes\mathbf{M}\,d\Theta\,d\Phi. \tag{2.12}$$

$\pi$-periodic von Mises distribution is then used for $\rho(\Theta)$ and $\rho(\Phi)$ [27],

$$\rho(\theta) = \frac{\exp(b\cos(2\theta))}{2\int_0^{\pi}\exp(b\cos(x))\,dx}, \tag{2.13}$$

in which $\theta$ is a variable representing $\Theta$ or $\Phi$, $b > 0$ is the concentration parameter, $\frac{1}{\pi}\int_0^{\pi}\exp(b\cos(x))dx$ is the modified Bessel function of the first kind of order zero.

From figure 4a,b, we can find that in-plane angle ($\Theta$) varies linearly from endocardium to epicardium for both RBM cases, but the fibres are much dispersed for case LDDMM, especially near the endocardium and epicardium, where myofibres align more longitudinally ($\mathbf{l}_0$). The out-of-plane angle ($\Phi$) is zero for both RBM cases since RBM generated myofibres only lie in the $\mathbf{c}_0 - \mathbf{l}_0$ plane. However, out-of-plane dispersion can be seen in case LDDMM shown in figure 4b. We now determine the in/out-of-plane dispersions from the angle differences between case LDDMM and RBM$^{uni}$. Figure 4c,d shows the histograms of in/out-of-plane dispersion in the LV, both $\Theta$ and $\Phi$ centre around 0°. The maximum-likelihood method *mle()* from Matlab is used to fit $\rho_{ip}$ and $\rho_{op}$ to the histograms of the in/out-of-plane dispersions, with $b_1 = 1.6153$ for the in-plane dispersion, and $b_2 = 1.2144$ for the out-of-plane dispersion.

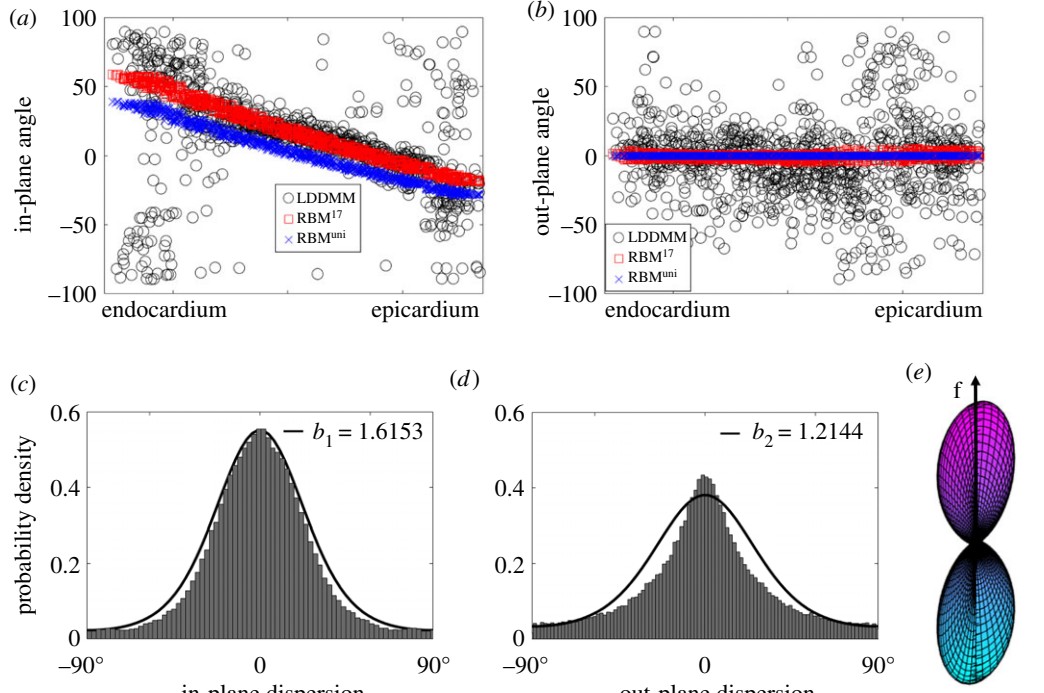

**Figure 4.** Fibre dispersion quantified from the DT-MRI dataset. (*a*) shows the in-plane angle $\Theta$ and (*b*) the out-of-plane angle $\Phi$ across the LV ventricular wall; (*c*) is the in-plane dispersion distribution with fitted $\rho(\Theta, b_1)$ and (*d*) is the out-of-plane dispersion distribution with fitted $\rho(\Phi, b_2)$; (*e*) a three-dimensional surface plot defined by the vector $\rho(\Theta, \Phi)\mathbf{f}(\Theta, \Phi)$ with $\rho(\Theta, \Phi) = \rho(\Theta, b_1)\rho(\Phi, b_2)$. The negative angle in (*a*) suggests the in-plane fibre vector lies in the fourth quadrant ($+\mathbf{c}_0$ and $-\mathbf{l}_0$), and similarly in (*b*) for the out-of-plane fibre vector, which lies in the fourth quadrant of plane ($-\mathbf{l}_0$ and $+\mathbf{r}_0$). All values are used for determining the in-plane and out-of-plane dispersions in (*c*) and (*d*).

Without loss of generality, we consider the mean fibre direction along $\mathbf{e}_3$, the sheet direction along $\mathbf{e}_1$ and the sheet-normal direction along $\mathbf{e}_2$. Then the in-plane distribution is $\rho_{\mathrm{ip}}(\Theta - 0, b_1)$, the out-of-plane distribution is $\rho_{\mathrm{op}}(\Phi - (\pi/2), b_2)$, and

$$
\begin{aligned}
\mathbf{H} &= \int_0^\pi \int_0^{2\pi} \frac{1}{N} \rho(\Theta, b_1)\, \rho(\Phi - \pi/2, b_2)\, \sin(\Theta)\, \mathbf{M} \otimes \mathbf{M}\, \mathrm{d}\Theta\, \mathrm{d}\Phi \\
&= \begin{pmatrix} 0.086 & & \\ & 0.268 & \\ & & 0.646 \end{pmatrix} \\
&= H_{11}\mathbf{s}_0 \otimes \mathbf{s}_0 + H_{22}\mathbf{n}_0 \otimes \mathbf{n}_0 + H_{33}\mathbf{f}_0 \otimes \mathbf{f}_0.
\end{aligned}
\tag{2.14}
$$

Similar to [18], we assume the active Cauchy stress with dispersed myofibres is

$$
\boldsymbol{\sigma}^a = T_a H_{11}\hat{\mathbf{s}} \otimes \hat{\mathbf{s}} + T_a H_{22}\hat{\mathbf{n}} \otimes \hat{\mathbf{n}} + T_a H_{33}\hat{\mathbf{f}} \otimes \hat{\mathbf{f}}.
\tag{2.15}
$$

Thus we have $n_{\mathrm{s}} = H_{11} = 0.086$, $n_{\mathrm{n}} = H_{22} = 0.268$ and $n_{\mathrm{f}} = H_{33} = 0.646$ for case RBM$^{\mathrm{uni}}$.

## 2.3. Boundary conditions and implementations

The bi-ventricular model is implemented using the nonlinear FE software Abaqus (Dassault Systemes, Johnston, RI, USA). In order to simulate diastolic filling and systolic ejection, a lumped model for the pulmonary and systemic circulation systems is attached to this bi-ventricular model, which is realized through a combination of surface-based fluid cavities and fluid exchanges [46] as shown in figure 5. We define the mass flow rate between two different cavities as

$$
\dot{m} = \rho \dot{V} A,
\tag{2.16}
$$

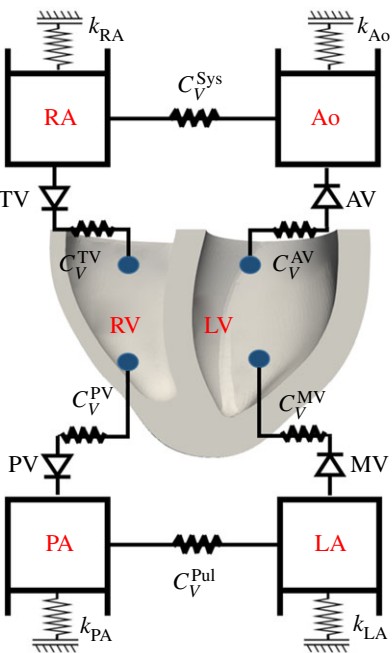

**Figure 5.** Schematic of the bi-ventricular model coupled with a circulatory system. MV, mitral valve; AV, aortic valve; RA, right atrium; TV, tricuspid valve; PV, pulmonary valve; LA, left atrium; RA, right atrium; Ao, aorta; Sys, systemic circulation; Pul, pulmonary circulation; and PA, pulmonary artery. Grounded spring with a stiffness ($k$) is tuned to provide the appropriate pressure–volume response (i.e. compliance) for that cavity. $C_V$ is viscous resistance coefficient to describe resistance between cavities. One-direction flow through valves is controlled by setting fluid exchanging properties between the cavities.

where $\rho$ is the blood density, $A$ is the effective area between the two connected cavities and $\dot{V}$ is the fluid flux. $\dot{m}$ is further related to the pressure difference

$$\Delta pA = C_V \dot{m} + C_H \dot{m}|\dot{m}|, \tag{2.17}$$

where $\Delta p$ is the pressure difference between two connected cavities, $C_V$ is viscous resistance coefficient, and $C_H$ is hydrodynamic resistance coefficient, and $C_H = 0$ in this study. This type of boundary conditions is equivalent to a simplified two-element Windkessel model. Parameters for the lumped circulation system are listed in table 2 and scaled from [6] by taking into account the dimensions of the neonatal porcine heart. For example, the total blood volume is around 80 ml for a newborn piglet [47], much less than in an adult porcine ($67.2 \pm 4.12$ ml kg$^{-1}$) [48], the valvular area in a newborn heart is about one-tenth of the area in an adult heart [49–51], and the diameter of blood vessel is also much smaller in the newborn piglet compared with an adult porcine [52], which suggests that under similar pressure loadings, the vessel compliance, calculated as $\Delta V/\Delta P$ will be much less in a newborn porcine because of much smaller $\Delta V$ in a newborn piglet.

Parameters for passive strain energy function and the maximum active tension from myocytes ($T_{max}$) are listed in table 3. Initial values for passive response are from [39], $a_i$ ($i \in \{1, 4f, 4n, 8fs, 8fn\}$) are further reduced by half together with chosen $T_{max}$ to ensure both LV and RV can achieve ejection fraction (EF) within the physiological range (EF > 50%). Values of $b_i$ ($i \in \{1, 4f, 4n, 8fs, 8fn\}$) are kept the same as in [39]. Note that because of missing measured data (wall motion, ventricular pressure) for the porcine heart, rather than constructing a personalized model [5,6], we only aim to obtain a set of parameters with which the bi-ventricle behaves physiologically.

The FE nodes on the top basal plane are constrained along the longitudinal axis but free to move within the basal plane. The longitudinal axis is defined as the line passing the LV basal centre and perpendicular to the basal plane. To start the simulation, linearly increased blood pressures from 0 to end-diastolic values are first applied to the inner surfaces of the bi-ventricular model, 8 mmHg in the LV and 4 mmHg in the RV. Typical diastolic pressures inside the pulmonary, left atrium, aorta and right atrium are also applied to those four cavities (10, 8, 67.5 and 4 mmHg [53]). Then the bi-ventricular model starts iso-volumetric contraction ($t = 0$ s), followed by systolic ejection when the ventricular pressure is higher than that of the aorta (around $t = 0.045$ s), and then the iso-volumetric relaxation. Systolic ejection ends at 0.12 s; 1 s is chosen for a whole cardiac cycle for computational

**Table 2.** Parameter values for the lumped circulatory model as shown in figure 5. $C_V$ is the viscous resistance coefficient, and $k$ is the stiffness of the grounded spring. Corresponding values for the equivalent Windkessel model is also listed for reference including the resistance ($R$) and the compliance ($C$). Note that the compliances of the RA and LA are not constant but varied to ensure constant end-diastolic pressure, which are not listed here.

| ABAQUS | | | Windkessel equivalent | | |
|---|---|---|---|---|---|
| name | value | unit | name | value | unit |
| $C_V^{AV}$ | 20.0 | $\mathrm{MPa \cdot mm^2 \cdot s\, tonne^{-1}}$ | $R_{AV}$ | 0.150 | $\mathrm{mmHg \cdot s\, ml^{-1}}$ |
| $C_V^{MV}$ | 50.0 | — | $R_{MV}$ | 0.375 | — |
| $C_V^{PV}$ | 55.0 | — | $R_{PV}$ | 0.412 | — |
| $C_V^{TV}$ | 16.0 | — | $R_{TV}$ | 0.120 | — |
| $C_V^{Sys}$ | 3600.0 | — | $R_{Sys}$ | 27.0 | — |
| $C_V^{Pul}$ | 300.0 | — | $R_{Pul}$ | 2.25 | — |
| $k_{Ao}$ | 0.8 | $\mathrm{N\, mm^{-1}}$ | $C_{Ao}$ | 0.061 | $\mathrm{ml\, mmHg^{-1}}$ |
| $k_{PA}$ | 0.8 | — | $C_{PA}$ | 0.065 | — |
| $k_{LA}$ | 0.1 | — | $C_{LA}$ | — |  |
| $k_{RA}$ | 0.1 | — | $C_{RA}$ | — |  |

**Table 3.** Parameter values for passive properties of the LV and RV myocardium.

|  | $a$ (kPa) | $b$ | $a_f$ (kPa) | $b_f$ | $a_n$ (kPa) | $b_n$ | $a_{fs}$ (kPa) | $b_{fs}$ | $a_{fn}$ (kPa) | $b_{fn}$ (kPa) | $T_{max}$ (kPa) |
|---|---|---|---|---|---|---|---|---|---|---|---|
| LV | 0.038 | 18.143 | 3.5335 | 1.339 | 1.373 | 4.495 | 0.929 | 4.067 | 1.771 | 8.225 | 180 |
| RV | 0.485 | 7.513 | 2.777 | 1.685 | 0.704 | 9.407 | 0.121 | 15.314 | 1.351 | 17.235 | 135 |

convenience. In order to ensure the end-diastolic pressures in both LV and RV are same at next cardiac cycles, end-diastolic pressures in both atria are maintained constant.

# 3. Results

We first compare the heart pump function for cases LDDMM, RBM[17] and RBM[uni] without cross-fibre active tension. We then analyse the effect of cross-fibre active tension in case RBM[uni]. Finally, we include dispersed active tension derived from DT-MRI myofibres in case RBM[uni] and compared with case LDDMM.

## 3.1. No cross-fibre active tension

Figure 6a shows the pressure–volume loops from the three cases with no cross-fibre active tension. Although they all have the same end-diastolic pressure, the LV end-diastolic volume from case LDDMM (2.87 ml) is slightly larger than the other two rule-based cases (2.83 ml), the relative difference is around 1.4%. The LV end-systolic volume in case LDDMM is also the smallest (1.38 ml). Interestingly, though myofibre structures in the RV for the three cases are same, however, due to the difference in LV dynamics, the RV end-systolic volume from case LDDMM is also the smallest (0.87 ml).

Figure 6b shows ejection fractions for the three cases. Again, case LDDMM achieves higher ejection fraction both at LV (51.92%) and RV (55.47%) than the two rule-based cases. Furthermore, the LV ejection fractions for cases RBM[17] and RBM[uni] are less than 50%, which are below literature reported normal range (50%–75%), indicating the LV pump function is suboptimal in those two cases.

Figure 6c shows the average end-systolic stress for the entire LV along the circumferential, radial and longitudinal directions, respectively. Although the circumferential stress from case LDDMM is lower

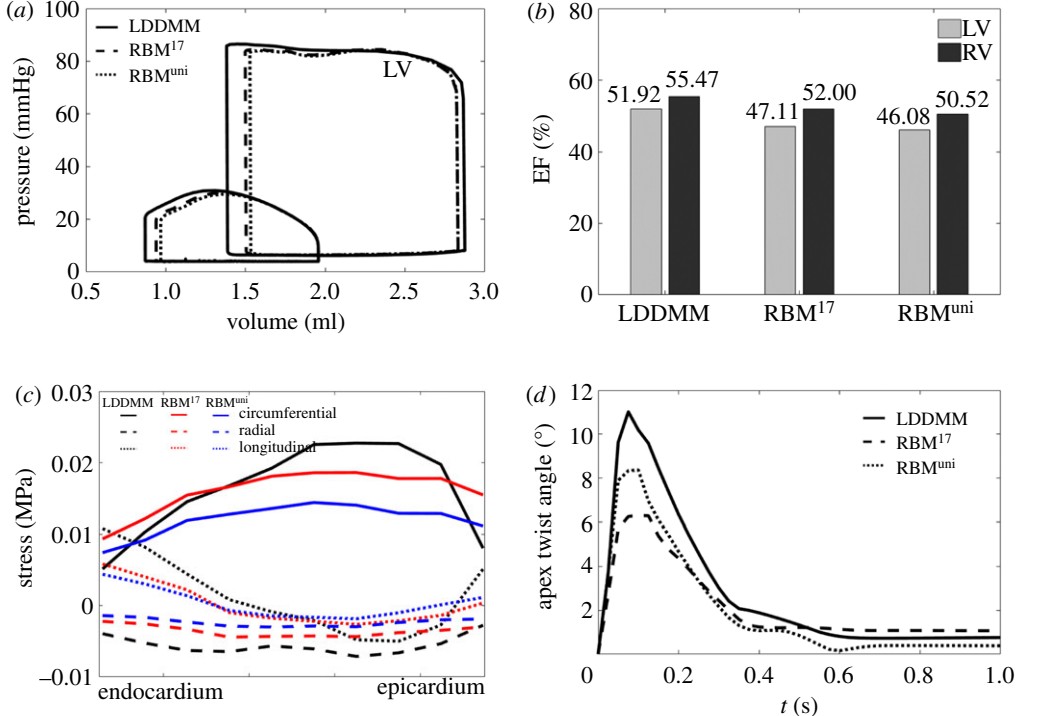

**Figure 6.** Simulated pump functions from cases LDDMM, RBM[17] and RBM[uni], including (*a*) Pressure–volume loops of LV and RV, (*b*) LV and RV ejection fractions, (*c*) stress distribution across the wall at end-systole and (*d*) apex twist angle.

near endocardium and epicardium than RBM cases, it is much higher in the midwall, with the lowest value from case RBM[uni]. Contrary to the circumferential stress, the longitudinal stress is higher in case LDDMM at endocardium and epicardium, while lowest at part of the midwall. The opposite trends of the circumferential and longitudinal stress levels in case LDDMM may compensate each other to achieve a deeper systolic contraction than cases RBM[17] and RBM[uni]. The radial stress is negative for all three cases with the lowest in case LDDMM.

Figure 6*d* is the apex twist angle within one cardiac cycle. The twist angle is defined as the rotation of the apex with respect to the basal plane at end-diastole. The apex from case LDDMM twists more compared with cases RBM[17] and RBM[uni], with a peak value of 11°, which is well within the reported ranges in healthy hearts ($10.2 \pm 7.6°$) [54]. Therefore, a more efficient pump function is achieved in case LDDMM compared with the RBM cases. Difference between the two rule-based cases are subtle, only slightly improved pump function can be found in case RBM[17], compared with case RBM[uni], but it has a reduced apex twist.

Figure 7*a*–*c* shows the end-systolic myofibre stress distributions for the three cases. In case LDDMM, higher myofibre stress ($\hat{\mathbf{f}} \cdot (\boldsymbol{\sigma}\hat{\mathbf{f}})$) can be found at both the endocardial and epicardial surfaces, especially in the LV side, while its distribution is less uniform compared with the two RBM cases. Figure 7*d*–*f* shows the strains along myofibre at end-systole. Strain distributions are similar in the two RBM cases, but the great difference is seen from the LDDMM case. The less uniform distributions of stress and strain in case LDDMM may be partially explained by much dispersed myofibre structures. The angle between the long-axis and the longitudinal axis at end-systole, defined in figure 7*d*–*f*, is largest in the LDDMM case (8.7°) and lowest in RMB[uni] (4.2°), also suggesting different deformed end-systolic shapes.

## 3.2. RBM[uni] with cross-fibre active tension

Based on case RBM[uni], five different sets of $n_s$ and $n_n$ are chosen to investigate how they affect ventricular dynamics. These are: (i) $n_s = 0$, $n_n = 0$, (ii) $n_s = 0.2$, $n_n = 0$, (iii) $n_s = 0.4$, $n_n = 0$, (iv) $n_s = 0.0$, $n_n = 0.2$ and (v) $n_s = 0.0$, $n_n = 0.4$. For all simulations $n_f = 1.0$. Figure 8 shows the pump functions with varied $n_s$ or $n_n$. If we only consider cross-fibre active tension along the sheet direction, then the pressure–volume loop enclosed area is reduced as shown in figure 8*a*, suggesting that the active tension along the sheet direction will counteract the myofibre contraction. On the other hand, non-zero $n_n$ increases the area enclosed by the pressure–volume loop and enhances the cardiac work. For example, with $n_s = 0.4$, the

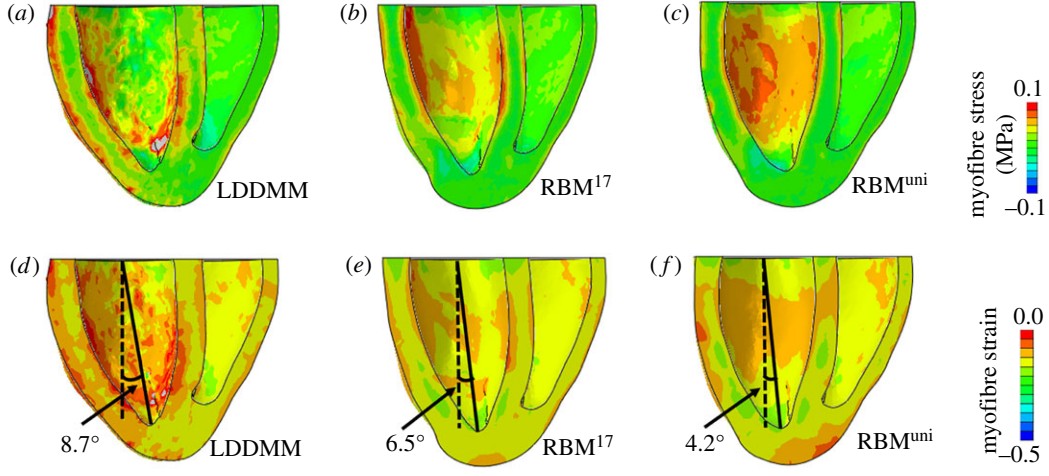

**Figure 7.** Myofibre stress and strain distributions at end-systole for cases LDDMM, RBM[17] and RBM[uni]. The solid lines in (*d–f*) are the long-axis which links the LV basal centre and the LV apex, and the longitudinal axis is represented by the dash line passing the LV basal centre and perpendicular to the basal plane.

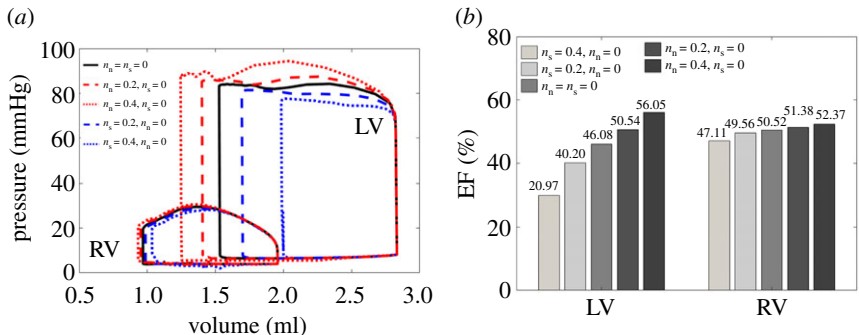

**Figure 8.** Pump functions with varied $n_s$ and $n_n$ in case RBM[uni]. $n_f = 1.0$ for all simulations. (*a*) Pressure–volume loops of LV and RV, and (*b*) ejection fractions for LV and RV.

LV EF is around 29.97%, which is much less than the case with $n_s = 0$ (46.08%), while with $n_n = 0.4$, LV EF is increased by 10% as shown in figure 8*b*. Therefore, active tension along the sheet-normal direction is beneficial to the pump function, but contraction along the sheet direction has the opposite effect.

Figure 9 shows results from case RBM[uni] with dispersed active contraction, modelled by the structural tensor from equation (2.14). In this case, case RBM[uni] has nearly the same LV P–V loop as case LDDMM, and the apical twist is also similar to case LDDMM (figure 9*a,b*). Only a small difference in end-diastolic volume ($\approx 1.4\%$) is observed between the two models. On the other hand, figure 9*c,d* shows that the end-systolic circumferential stress is much lower compared with case LDDMM, particularly in the midwall. The longitudinal and radial stresses are also slightly higher in the midwall because of non-zero $n_n$ and $n_s$.

In summary, compared with case LDDMM, case RBM[uni] shows a lower and more homogeneous stress level but achieves a similar pump function if using a suitable general structural tensor approach for the cross-fibre contraction.

It is interesting to see if similar results could be obtained without any knowledge of the patient-specific fibre field. To this end, we run extra simulations based on RBM[uni] using literature-based values for $n_f$, $n_s$ and $n_n$. Specifically, we consider (i) no dispersion $n_f = 1$, $n_s = n_n = 0$, (ii) $n_f = 0.879$, $n_s = 0.009$, $n_n = 0.112$ [45] and (iii) $n_f = 0.646$, $n_s = 0.086$, $n_n = 0.268$, derived from DT-MRI in this study. The fibre rotation angles are also chosen from 30° to −30° (exRBM[1]), 45° to −45° (exRBM[2]), or 60° to −60° (exRBM[3]) [15]. The results are summarized in figure 10 in terms of the LV and RV ejection fractions. Clearly, EFs increase with fibre rotation angles, as more myofibres align longitudinally which enhance the active contraction. Different dispersion parameters also affect the pump function. Compared with case LDDMM, the EFs are lower in exRBM[1] (39.37% (LV), 45.89% (RV)), and still lower in exRBM[2] (47.86% (LV), 51.72% (RV)). Only exRBM[3] with DT-MRI-derived dispersion parameters can achieve

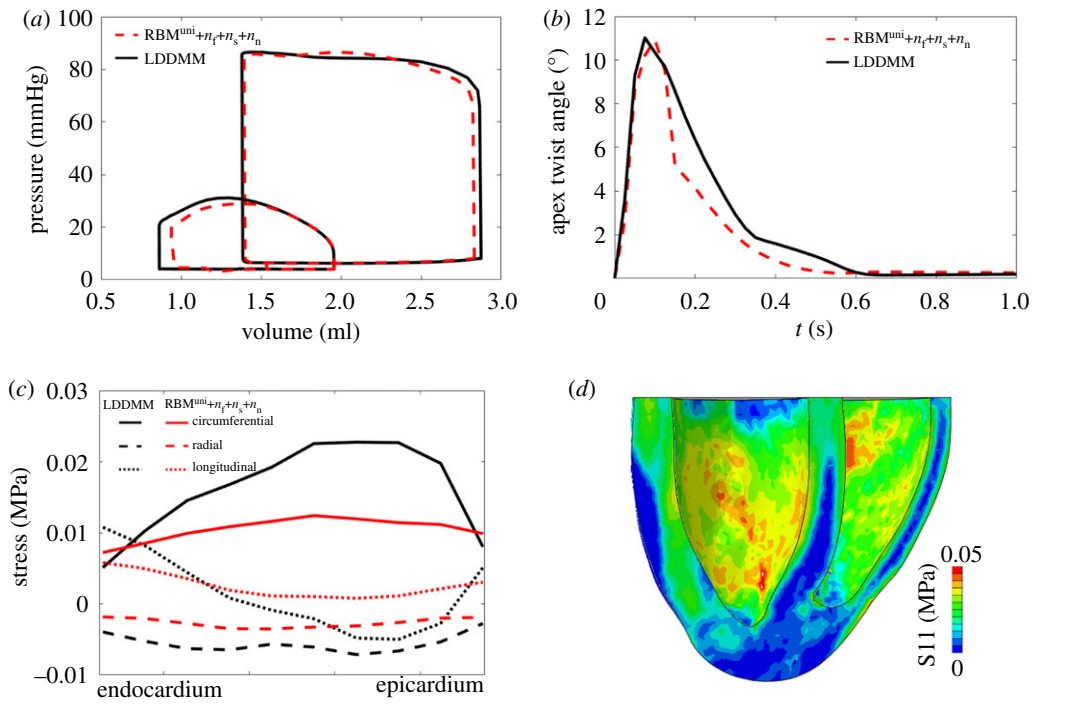

**Figure 9.** Pump function comparisons between case LDDMM and case RBM$^{uni}$ with cross-fibre contraction. (a) pressure–volume loops, (b) apex twist angle, (c) intramural stress across the entire LV wall and (d) myofibre stress distribution from case RBM$^{uni}$ with cross-fibre contraction at end-systole.

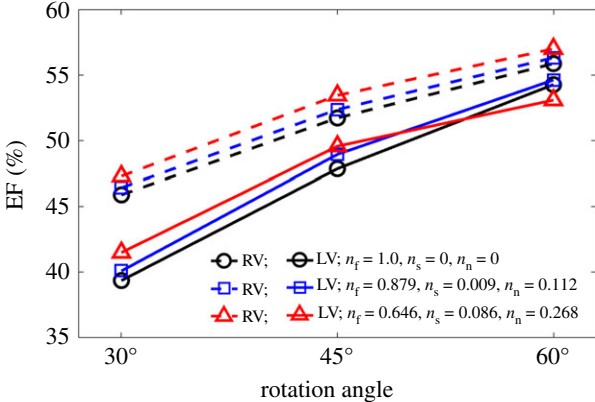

**Figure 10.** Predicted ejection fractions with literature-based myofibre rotation angles [15] and dispersion parameters [45] using case RBM$^{uni}$. The results are to be compared with the LDDMM case in figure 7b, which has the mean fibre rotation angles 40° to −30°, and EFs of 51.92% (LV) and 55.47% (RV).

the similar pump functions as in case LDDMM, though the myofibre rotation angles (60° to −60°) are much greater than case LDDMM (mean angles 40° to −30°). This would suggest that subject-specific myofibre structure is necessary for cardiac mechanic modelling, as using literature-based myofibre structures seems to underestimate the pump function.

## 4. Discussion

In this study, LDDMM-based *Deformetrica* [41] is used to warp a canine bi-ventricle to a neonatal porcine heart, and DT-MRI-measured myofibre structure is then mapped to a porcine heart by solving a Laplace system. Base on the mapped DT-MRI measured myofibre structure, two simplified fibres are further generated using a rule-based approach. Our results show that under the same pre-/after-loading conditions, both LV and RV have a higher pump function in the case with LDDMM-mapped fibres

compared with the rule-based cases, while case LDDMM experiences higher myofibre stress and more heterogeneous stress pattern than rule-based cases. Large differences can be expected when using literature-based fibre structures and dispersion parameters compared with case LDDMM. Those different results highlight the necessity of use realistic myofibre structure for personalized cardiac modelling as demonstrated in other studies [15,20,21,24].

In case LDDMM, the high active fibre stresses at both epicardial and endocardial surfaces (figure 7a) can potentially enhance the long-axis shortening and also apical twist (figure 6d). In fact, long-axis shortening in systole with respect to end-diastole is slightly higher in case LDDMM (−7.3%) than other two cases (−6.8% for RBM[17] and −7% for RBM[uni]). Our results show (figure 4a,b) that DT-MRI-derived myofibres do not lie in the $c_0 − l_0$ plane but are dispersed. Thus the active tension in case LDDMM is generated along fibres dispersed with both in-plane and out-of-plane components. In §2.2.3, we firstly quantify myofibre dispersion with in-plane and out-of-plane distributions, and then introduce a structural tensor $\mathbf{H}$ [26,55] by fitting to the measured in/out-of-plane dispersions. The $\pi$-period von Mises distribution is used to describe myofibre dispersion, good agreement can be achieved as shown in figure 4c,d. While it may not be guaranteed that the von Mises distribution can be applied to pathological tissues, such as myocardial infarction [56].

To take into account myofibre dispersion in the active stress formulation, we further assume the generated active force $T_a$ is dispersed along the fibre, the sheet and the sheet-normal directions with proportions determined by $\mathbf{H}$. We find that cross-fibre contraction is highest along the sheet-normal direction compared with that of the sheet direction, but much lower than along mean fibre direction. Furthermore, active contraction in the sheet-normal direction can facilitate contraction, but not in sheet direction. This is because myofibres dominantly lie in $c_0 − l_0$ plane, in which $\mathbf{f}$ and $\mathbf{n}$ are defined, and contraction along $\mathbf{f}$ and $\mathbf{n}$ causes circumferential and long-axial shortening [57], so the wall thickens to maintain the constant wall volume if the material is incompressible. While transmural contraction along $\mathbf{s}$ causes wall thinning, which counteracts myofibre contraction. Note that in this study, the sheet direction is defined transmurally across the wall, which is consistent with studies from [7,11,18], though some studies define it as the sheet-normal direction [6]. Unlike the myofibres which rotate from endocardium to epicardium, here the sheet direction is assumed to align the radial direction in all cases. In other words, the sheet rotation angle is chosen to be zero. To evaluate this assumption, we have tested three sets of sheet rotation angles as in [15]: 30° to −30°, 45° to −45° and 60° to −60°, based on case RBM[uni] with dispersed active tension. The results show that the sheet rotation angle has little effect on ventricular pump function, and the differences in ejection fraction between different sheet rotation angles are within 1%. This agrees with observations from other groups. For example, Wang et al. [15] found that the sheet rotation angle nearly has no influence on passive mechanics in an LV model.

We now compare our values of cross-fibre proportions ($n_s = 0.086$, $n_n = 0.268$ and $n_f = 0.646$) with previous studies. Based on the experimental study by Lin & Yin [38], Guccione and co-workers introduced cross-fibre active contraction with $n_s = 0.0$, $n_n = 0.4$ and $n_f = 1.0$ [37]. In a recent study, Sack et al. [6] inversely determined cross-fibre contraction ratios[5] in a healthy porcine heart ($n_n = 0.07$) and a failure heart ($n_n = 0.14$) with $n_f = 1.0$ and $n_s = 0$. In our study, $n_n$ (0.268) is higher than that of Sack et al.'s study [6]. This could be due to (i) subject variation; (ii) higher $n_f = 1.0$ used in their study (our $n_f = 0.646$), leading to a higher contraction along the averaged myofibre direction so a lower $n_n$ could match the measured pump function; (iii) they inversely determined $n_n$ and $T_a$, which are not from measurements. In this study, proportions of cross-fibre contraction are derived directly from intrinsic fibre structures, which have a clear biological explanation. When normalized by $n_f$, the ratio between the sheet-normal and myofibre direction is 41%, which agrees with the ratio reported by Lin & Yin (40%) [38]. We further calculate the dispersion parameters from a recent study on neonatal porcine heart by Ahmad et al. [25], $n_f = 0.68$ and $n_n = 0.32$ with nearly negligible $n_s \approx 0.0009$, again very close to our values in this study. We are not aware of any available experimental measurements for estimating $n_n$ and $n_s$ in the myocardium.

Rodríguez-Cantano et al. [24] argued that RBM tends to exaggerate myofibre-layered architecture and the passive stiffness of the ventricle, while DT-MRI-measured fibres may underestimate ventricular stiffness due to measurement noise and uncertainties. We find that when taking into account the cross-fibre contraction in the case RBM[uni], we can achieve similar systolic contraction as case LDDMM (figure 9) with less heterogeneous stress patterns. Because of challenging of in vivo DT-MRI acquisition, rule-based myofibre structures will continue to be used when modelling cardiac mechanics, even in personalized models. Our results suggest by incorporating fibre dispersion using a structural tensor, RBM-based model can be a good approximation of the most realistic myofibre

---

[5]Note that in Sack et al.'s work [6] they used notation $n_s$ for $n_n$ due to a different definition.

structure as derived from DT-MRI, and the structural tensor may be determined either from limited *in/ex vivo* DT-MRI data [58] or inversely estimated, while caution needs to be paid when myofibre structures are from different subjects or species. There is a small difference (around 1.4%) in end-diastolic volume in figure 9*a*, presumably because the dispersion is not included in the passive constitutive law. Given that exclusion of compressed fibres using structural tensor approach is non-trivial in the passive modelling [59,60], we will leave the work of including dispersion in the passive model in future.

Using material parameters estimated from *ex vivo* measurements to describe *in vivo* material behaviours is a standing challenge. Published studies have suggested passive parameters estimated from *ex vivo* experiments can overestimate the stiffness *in vivo* [15,33,61]. Hence, most of the studies, ours included, scaled the parameters from *ex vivo* data to match the *in vivo* dynamics [5,6,61]. Here, the initial passive parameters are adopted from our previous study [39] which were inferred from *ex vivo* neonatal myocardial stretching experiments [40]; then $a$, $a_f$, $a_n$, $a_{fs}$ and $a_{fn}$ are scaled to achieve the targeted end-diastolic volumes. The myocardial contractility $T_{max}$ is determined by matching the targeted ejection fractions (greater than 50%) for both the LV and RV. We further assume the passive scaling factor is the same for the LV and RV. Thus only three parameters need to be determined: the passive scaling factor, $T_{max}$ for the LV and $T_{max}$ for the RV. The sensitivity study on the passive parameters and $T_{max}$, and an illustration of their inferences are provided in the electronic supplementary material.

The convexity of the HO type strain energy function requires all parameters greater than zero as suggested in [7], which is satisfied in our approach. However, as pointed out by Giantesio *et al.* [62], the polyconvexity of the total energy function (passive and active) may not be ensured even though the passive strain energy function is convex. Although we have not experienced stability issues using the active stress approach, we must point out this approach may not be thermodynamically consistent. For generalized thermodynamically consistent approaches, the reader is referred to [9,34,62].

Due to lack of DT-MRI data for the RV from the canine experiment, a rule-based approach is used for generating fibre structure in the RV, and zero cross-fibre contraction is assumed. This can be readily improved if measured RV fibre structure becomes available. We note there is a difference in the RV systolic function even though the RV model is identical in all three cases. In particular, the RV contracts more in case LDDMM than in the two RBM cases. We think this is due to the different LV contraction in the three cases. For instance, the end-systole angle between the long-axis and longitudinal axis is different in each case. Palit *et al.* [19] also found that there are strong interactions between the LV and RV dynamics in diastole. This highlights the importance of LV-RV interaction on cardiac pump function, which is why the bi-ventricle model is used. In addition, the LDDMM framework [41] relies on geometrical features for warping the two different geometries, a bi-ventricular model has much richer information compared with a stand-alone LV model, in particular in the RV-LV insertion regions.

It is expected that there are differences in myofibre structure between the porcine heart and the canine heart, but this is difficult to assess as we do not have measured DT-MRI fibre structure for the porcine heart. However, despite the species difference, we find that the mapped canine myofibre structure agrees well with other studies in terms of mean values [6,23,25] (table 1). For instance, Ahmad *et al.* [25] measured myofibre rotation angles in LV free wall of neonatal hearts (anterior $51.1 \pm 3.8°$ to $-51.1 \pm 3.8°$ and posterior $40.2 \pm 2.9°$ to $-40.2 \pm 2.9°$). Sack *et al.* [6] reported fibre rotation angles for a normal adult porcine heart based on DT-MRI measurements (endocardium: $66.5 \pm 16.6°$, epicardium: $-37.4 \pm 22.4°$). Myofibre rotation angles from published experimental and numerical studies are also summarized in the electronic supplementary material.

The spatial variations of the material properties have not been considered in this study, and the same averaged dispersed active contraction model is applied across the whole LV for case RBM$^{uni}$. This approximation may be reasonable for healthy hearts, but questionable for pathological cases. For example, the myocardium is known to be more heterogeneous post-myocardial infarction [56].

Finally, we would like to mention other limitations of our study. In the boundary conditions we used, the basal plane of the models is constrained along the longitudinal direction, and the rest nodes in the basal plane are free to move. This type of boundary conditions does not represent *in vivo* conditions due to the lack of the pericardium and great vessels. Under *in vivo* situation, with the constraints imposed by the pericardium, the apex does not move much. Instead, the basal plane moves downward towards the apex in systole and moves upward in diastole. In a recent study, Pfaller *et al.* [63] demonstrated that simulated cardiac mechanics could be much closer to the measured heart motion by including the pericardium influences, which highlights the necessity of pericardial–myocardial interaction. A simplified lumped circulation model is used to provide pressure boundary conditions, which is a simplification of pulmonary and systemic circulations. Coupling to a more realistic circulation model, such as one-dimensional systemic models [64,65], will allow us to simulate

more detailed cardiovascular function in pathological situations [66]. Furthermore, we have not coupled the blood flow inside ventricle, only applied a spatially homogeneous pressure to the endocardial surface, nor have we considered contraction delay due to the action potential propagation [9]. Tremendous efforts will be needed to address all those limitations, which is beyond the scope of this study.

# 5. Conclusion

In this study, we have developed a bi-ventricular porcine heart computational model from a neonatal dataset, with mapped myofibre architecture from an *ex vivo* canine DT-MRI dataset using an LDDMM framework. Different approximations of myofibre architecture based on widely used rule-based approaches are analysed in terms of cardiac pump function. Our results show that using DT-MRI derived myofibre architecture can enhance cardiac work, achieve higher ejection fraction and larger apical twist compared with rule-based myofibre models, even though they are all derived from the same DT-MRI dataset. Our work shows that the major difference between the LDDMM and RMB approaches is due to the fibre dispersion, which enables cross-fibre active tensions. These are not captured by standard RBM-based models. Introducing regional dependent fibre structure in RBM is not sufficient to improve the model. However, when the myofibre dispersion is taken into consideration, a simplified RBM-based cardiac model can achieve similar pump function as the LDDMM-based model. We further note that in RBM-based cardiac models, the cross-fibre active tension along the sheet-normal direction can enhance active contraction, but the opposite is true along the sheet direction.

Data accessibility. Electronic supplementary material is provided. The datasets supporting this article have been uploaded to github as part of the electronic supplementary material, https://github.com/HaoGao/FibreGeneration-LDDMM, and the DOI is 10.5281/zenodo.3458254.

Authors' contributions. D.G. performed the modelling, analysed results and drafted the manuscript. J.Y. contributed to the model development. H.G. and X.L. conceived the study, analysed the results and supervised the project. H.G. coordinated the study. All authors contributed to the writing of the manuscript and gave final approval for publication.

Competing interests. Prof. Xiaoyu Luo is a member of the board of Royal Society Open Science at the time of submission.

Funding. We are grateful for the funding provided by the UK EPSRC (grant no. EP/N014642/1). D.G. also acknowledges funding from the Chinese Scholarship Council and the fee waiver from the University of Glasgow. H.G. also thanks for the funding from the National Natural Science Foundation of China (grant no. 11871399). Thanks also extend to all members of the Living Heart Project.

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
