## [Reviewer comments · Royal Society Open Science]

Review History

RSOS-191655.R0 (Original submission)

Review form: Reviewer 1

Is the manuscript scientifically sound in its present form?

Yes

Are the interpretations and conclusions justified by the results?

Yes

Is the language acceptable?

Yes

Do you have any ethical concerns with this paper?

No

Have you any concerns about statistical analyses in this paper?

No

Recommendation?

Accept with minor revision (please list in comments)

Comments to the Author(s)

Comments for the reviewer are attached (Appendix A).

Review form: Reviewer 2

Is the manuscript scientifically sound in its present form?

No

Are the interpretations and conclusions justified by the results?

Yes

Is the language acceptable?

Yes

Do you have any ethical concerns with this paper?

No

Have you any concerns about statistical analyses in this paper?

No

Recommendation?

Major revision is needed (please make suggestions in comments)

Comments to the Author(s)

The manuscript deals with an important problem related to imaging-based computational models of biological tissues, with particular reference to the cardiac wall and the myocardial fibres distribution.

The paper is well written, and I appreciate it, though I think several weak points need to be addressed before suggesting its publication.

Major Concerns:

Authors refer to Appendices I do not find in the electronic PDF file. If they are Appendices, why using supplemental material?

The overview of the literature needs to be extended and commented adequately in terms of distributed material model, both passive and active.

First, it is well-known that the original fibre distributed model from Gasser et al. suffers several limitations, as shown in many scientific contributions, see e.g. [1-5]. Second, yet in a generalised electro-mechanical framework, several contributions have been proposed accounting both for fibre distribution and multiphysics couplings, see e.g. [6-10]. Third, generalised statistical descriptions of fibre-reinforced biological tissues, accounting for local (in-the-thickness) variation of the microstructures reinforcement have been proposed in the literature, see e.g. [11,12]. A proper description of these studies is in combination with the following comments on the study limitations.

Concerning the adopted active stress electro-mechanical constitutive model, I emphasise once again the scientific contributions already present in the literature, see also [13-17] other than the previous ones. My main concern here is the thermodynamical consistency of the chosen

constitutive law. In particular, the orthotropic description assumed should be corroborated by a proper strain energy density to be derived consistently. In the active stress case, this is probably not the case. Accordingly, the Authors must clearly indicate this limitation and refer to generalised thermodynamically consistent approaches.

The assumption of zero cross-fibre contraction for RV is questionable. Why do the authors do not make a parametric analysis for this choice, checking out if their results are robust against it?

In order to provide a more coherent picture of the statistical distribution modelled, I suggest the Authors use three-dimensional plots in-plane and out-of-plane as in [12]. The reader would benefit from such a description. Besides, the reproducibility of the constitutive choice would increase. In this case, I suggest the Authors identify the limitations associated with the “average” value of the probability distribution function implied by the Holzapfel et al. approach. It is well-known that for pathological tissues, in which a high value of variance characterises the distribution of the fibres, such an approach is misleading.

Substantial limitations in terms of chosen boundary conditions need to be discussed in detail, see e.g. [18].

Another major concern to me is the procedure adopted to select material parameters. Due to the high number of parameters to set, it is questionable if such a method returns:

- 1) well represented energy (polyconvexity);
- 2) robustness against slight variations in the parameters (which is always the case for biological materials);
- 3) close-to-uniqueness (stability) of the model.

Can the Authors provide some convergence analyses in these regards?

As a final limitation, I suggest the Authors comment extensively on the mismatch between the chosen geometry (porcine) and the chosen fibres dataset (dog). In these regards, it is well-known the close relation between structure and function. Therefore, a clear description of the missing features limiting the generality of the results is strongly required, though a specific software is used for the mapping. Similarly, the angle definition at endocardium and epicardium is referred to the AHA which is a standard adopted for human hearts. All of these different features make the model complex to reproduce. See e.g. [19-21].

Finally, I suggest an extensive parametric analysis in which the chosen material parameters lead to specific behaviours.

Minor Concerns:

Notation between small and capital letters for statistical variables has to be appropriately introduced.

Ref 19: missing volume

Ref 24: missing year

etc.

Line 10: heart[1,3] space

Partial List of suggested references:

[1] Federico et al. *Journal of biomechanics* 38 (10), 2008-2018

<https://doi.org/10.1016/j.jbiomech.2004.09.020>

[2] Pandolfi and Vasta *Mechanics of Materials* Volume 44, January 2012, Pages 151-162

<https://doi.org/10.1016/j.mechmat.2011.06.004>

[3] Vasta et al. *Probabilistic Engineering Mechanics* 37 (2014) 170-179

<http://dx.doi.org/10.1016/j.probengmech.2014.05.003>

[4] Gizzi et al. *Mechanics of Materials* 92 (2016) 119-138

<http://dx.doi.org/10.1016/j.mechmat.2015.09.008>

- [5] Gilchrist et al. *Meccanica* 52 (14), 3417-3429 <https://doi.org/10.1007/s11012-017-0646-9>
- [6] Gizzi et al. *Commun. Comput. Phys.* Vol. 17, No. 1, pp. 93-126 oi: 10.4208/cicp.091213.260614a
- [7] Pandolfi et al. *Journal of Biomechanics* 49 (2016) 2436-2444
<http://dx.doi.org/10.1016/j.jbiomech.2016.01.038>
- [8] Pandolfi et al. *Meccanica* DOI 10.1007/s11012-017-0622-4
- [9] Ruiz-Baier et al. *Commun. Comput. Phys.* Vol. 27, No. 1, pp. 87-115 doi: 10.4208/cicp.OA-2018-0253
- [10] Propp et a. *Biomechanics and Modeling in Mechanobiology* <https://doi.org/10.1007/s10237-019-01237-y>
- [11] Vasta et al. *International Journal of Non-Linear Mechanics* 106 (2018) 258-265
- [12] Gizzi et al. *J Eng Math* (2018) 109:211-226 <https://doi.org/10.1007/s10665-017-9943-5>
- [13] Cherubini et al. *Prog Biophys Mol Biol.* 2008 Jun-Jul;97(2-3):562-73. doi: 10.1016/j.pbiomolbio.2008.02.001.
- [14] Ruiz-Baier et al. *Mathematical Medicine and Biology* (2014) 31, 259-283
doi:10.1093/imammb/dqt009
- [15] Gantesio et al. *Journal of Elasticity*, Volume 137, Issue 1, pp 63-82
<https://doi.org/10.1007/s10659-018-9708-z>
- [16] Quarteroni et al. *Computer Methods in Applied Mechanics and Engineering* 314, 345-407
<https://doi.org/10.1016/j.cma.2016.05.031>
- [17] Barbarotta et al. <https://doi.org/10.1002/cnm.3137>
- [18] Pfaller et al. *Biomechanics and Modeling in Mechanobiology* (2019) 18:503-529
<https://doi.org/10.1007/s10237-018-1098-4>
- [19] Nasopoulou et al. https://doi.org/10.1007/978-3-030-21949-9_34
- [20] Capilnasiu et al. <https://doi.org/10.1007/s10237-018-1072-1>
- [21] Marlevi et al. doi:10.1038/s41598-018-37714-0

Decision letter (RSOS-191655.R0)

19-Dec-2019

Dear Dr Gao,

The editors assigned to your paper ("Effect of Myofibre Architecture on Ventricular Pump Function by Using a Neonatal Porcine Heart Model: from DT-MRI to Rule-based Methods") have now received comments from reviewers. We would like you to revise your paper in accordance with the referee and Associate Editor suggestions which can be found below (not including confidential reports to the Editor). Please note this decision does not guarantee eventual acceptance.

Please submit a copy of your revised paper before 11-Jan-2020. Please note that the revision deadline will expire at 00.00am on this date. If we do not hear from you within this time then it will be assumed that the paper has been withdrawn. In exceptional circumstances, extensions may be possible if agreed with the Editorial Office in advance. We do not allow multiple rounds of revision so we urge you to make every effort to fully address all of the comments at this stage. If deemed necessary by the Editors, your manuscript will be sent back to one or more of the original reviewers for assessment. If the original reviewers are not available, we may invite new reviewers.

- Data accessibility

If you wish to submit your supporting data or code to Dryad (<http://datadryad.org/>), or modify your current submission to dryad, please use the following link:
<http://datadryad.org/submit?journalID=RSOS&manu=RSOS-191655>

- Competing interests

- Authors' contributions

- Acknowledgements

- Funding statement

Best regards,

on behalf of Dr Dirk Drasdo (Associate Editor) and R. Kerry Rowe (Subject Editor)
openscience@royalsociety.org

Reviewers' Comments to Author:

Reviewer: 1
Comments to the Author(s)

Comments for the reviewer are attached.

Reviewer: 2
Comments to the Author(s)

The manuscript deals with an important problem related to imaging-based computational models of biological tissues, with particular reference to the cardiac wall and the myocardial fibres distribution.

The paper is well written, and I appreciate it, though I think several weak points need to be addressed before suggesting its publication.

Major Concerns:

Authors refer to Appendices I do not find in the electronic PDF file. If they are Appendices, why using supplemental material?

The overview of the literature needs to be extended and commented adequately in terms of distributed material model, both passive and active.

First, it is well-known that the original fibre distributed model from Gasser et al. suffers several limitations, as shown in many scientific contributions, see e.g. [1-5]. Second, yet in a generalised electro-mechanical framework, several contributions have been proposed accounting both for fibre distribution and multiphysics couplings, see e.g. [6-10]. Third, generalised statistical descriptions of fibre-reinforced biological tissues, accounting for local (in-the-thickness) variation of the microstructures reinforcement have been proposed in the literature, see e.g. [11,12]. A proper description of these studies is in combination with the following comments on the study limitations.

Concerning the adopted active stress electro-mechanical constitutive model, I emphasise once again the scientific contributions already present in the literature, see also [13-17] other than the

previous ones. My main concern here is the thermodynamical consistency of the chosen constitutive law. In particular, the orthotropic description assumed should be corroborated by a proper strain energy density to be derived consistently. In the active stress case, this is probably not the case. Accordingly, the Authors must clearly indicate this limitation and refer to generalised thermodynamically consistent approaches.

The assumption of zero cross-fibre contraction for RV is questionable. Why do the authors do not make a parametric analysis for this choice, checking out if their results are robust against it?

In order to provide a more coherent picture of the statistical distribution modelled, I suggest the Authors use three-dimensional plots in-plane and out-of-plane as in [12]. The reader would benefit from such a description. Besides, the reproducibility of the constitutive choice would increase. In this case, I suggest the Authors identify the limitations associated with the “average” value of the probability distribution function implied by the Holzapfel et al. approach. It is well-known that for pathological tissues, in which a high value of variance characterises the distribution of the fibres, such an approach is misleading.

Substantial limitations in terms of chosen boundary conditions need to be discussed in detail, see e.g. [18].

Another major concern to me is the procedure adopted to select material parameters. Due to the high number of parameters to set, it is questionable if such a method returns:

- 1) well represented energy (polyconvexity);
- 2) robustness against slight variations in the parameters (which is always the case for biological materials);
- 3) close-to-uniqueness (stability) of the model.

Can the Authors provide some convergence analyses in these regards?

As a final limitation, I suggest the Authors comment extensively on the mismatch between the chosen geometry (porcine) and the chosen fibres dataset (dog). In these regards, it is well-known the close relation between structure and function. Therefore, a clear description of the missing features limiting the generality of the results is strongly required, though a specific software is used for the mapping. Similarly, the angle definition at endocardium and epicardium is referred to the AHA which is a standard adopted for human hearts. All of these different features make the model complex to reproduce. See e.g. [19-21].

Finally, I suggest an extensive parametric analysis in which the chosen material parameters lead to specific behaviours.

Minor Concerns:

Notation between small and capital letters for statistical variables has to be appropriately introduced.

Ref 19: missing volume

Ref 24: missing year

etc.

Line 10: heart[1,3] space

Partial List of suggested references:

[1] Federico et al. Journal of biomechanics 38 (10), 2008-2018

<https://doi.org/10.1016/j.jbiomech.2004.09.020>

[2] Pandolfi and Vasta Mechanics of Materials Volume 44, January 2012, Pages 151-162

<https://doi.org/10.1016/j.mechmat.2011.06.004>

[3] Vasta et al. Probabilistic Engineering Mechanics 37 (2014) 170-179

<http://dx.doi.org/10.1016/j.probenmech.2014.05.003>

- [4] Gizzi et al. *Mechanics of Materials* 92 (2016) 119–138
<http://dx.doi.org/10.1016/j.mechmat.2015.09.008>
- [5] Gilchrist et al. *Meccanica* 52 (14), 3417–3429 <https://doi.org/10.1007/s11012-017-0646-9>
- [6] Gizzi et al. *Commun. Comput. Phys.* Vol. 17, No. 1, pp. 93–126 oi: 10.4208/cicp.091213.260614a
- [7] Pandolfi et al. *Journal of Biomechanics* 49 (2016) 2436–2444
<http://dx.doi.org/10.1016/j.jbiomech.2016.01.038>
- [8] Pandolfi et al. *Meccanica* DOI 10.1007/s11012-017-0622-4
- [9] Ruiz-Baier et al. *Commun. Comput. Phys.* Vol. 27, No. 1, pp. 87–115 doi: 10.4208/cicp.OA-2018-0253
- [10] Propp et a. *Biomechanics and Modeling in Mechanobiology* <https://doi.org/10.1007/s10237-019-01237-y>
- [11] Vasta et al. *International Journal of Non-Linear Mechanics* 106 (2018) 258–265
- [12] Gizzi et al. *J Eng Math* (2018) 109:211–226 <https://doi.org/10.1007/s10665-017-9943-5>
- [13] Cherubini et al. *Prog Biophys Mol Biol.* 2008 Jun-Jul;97(2-3):562-73. doi: 10.1016/j.pbiomolbio.2008.02.001.
- [14] Ruiz-Baier et al. *Mathematical Medicine and Biology* (2014) 31, 259–283
 doi:10.1093/imammb/dqt009
- [15] Giantesio et al. *Journal of Elasticity*, Volume 137, Issue 1, pp 63–82
<https://doi.org/10.1007/s10659-018-9708-z>
- [16] Quarteroni et al. *Computer Methods in Applied Mechanics and Engineering* 314, 345–407
<https://doi.org/10.1016/j.cma.2016.05.031>
- [17] Barbarotta et al. <https://doi.org/10.1002/cnm.3137>
- [18] Pfaller et al. *Biomechanics and Modeling in Mechanobiology* (2019) 18:503–529
<https://doi.org/10.1007/s10237-018-1098-4>
- [19] Nasopoulou et al. https://doi.org/10.1007/978-3-030-21949-9_34
- [20] Capilnasiu et al. <https://doi.org/10.1007/s10237-018-1072-1>
- [21] Marlevi et al. doi:10.1038/s41598-018-37714-

Author's Response to Decision Letter for (RSOS-191655.R0)

See Appendix B.

Decision letter (RSOS-191655.R1)

26-Feb-2020

Dear Dr Gao,

It is a pleasure to accept your manuscript entitled "Effect of Myofibre Architecture on Ventricular Pump Function by Using a Neonatal Porcine Heart Model: from DT-MRI to Rule-based Methods" in its current form for publication in *Royal Society Open Science*. The comments of the reviewer(s) who reviewed your manuscript are included at the foot of this letter.

You can expect to receive a proof of your article in the near future. Please contact the editorial

office (openscience_proofs@royalsociety.org) and the production office (openscience@royalsociety.org) to let us know if you are likely to be away from e-mail contact -- if you are going to be away, please nominate a co-author (if available) to manage the proofing process, and ensure they are copied into your email to the journal.

Kind regards,

Anita Kristiansen
Editorial Coordinator

on behalf of Dr Dirk Drasdo (Associate Editor) and R. Kerry Rowe (Subject Editor)
openscience@royalsociety.org

Associate Editor Comments to Author (Dr Dirk Drasdo):

Comments to the Author:

The authors have very constructively responded on each of the reviewers' comments and improved their manuscript. I must say I am impressed by the positive interaction between reviewer and authors.

I think this manuscript is ready for acceptance.

Appendix A

Effect of Myofibre Architecture on Ventricular Pump Function by Using a Neonatal Porcine Heart Model: from DT-MRI to Rule-based Methods

Overview:

This study investigates the differences in ventricular pump function as well as left ventricular end-systolic stresses and strains due to using fibres measured from DT-MRI versus rule-based methods (RBM) in a biventricular model. In particular, the authors illustrate how incorporating a non-symmetrical dispersed active tension model with their RBM fibre field results in pressure-volume loops (for LV and RV) which are similar to those obtained using fibres from DT-MRI. Additionally, the resulting angle of apical tilt is also more similar when incorporating the non-symmetrical dispersion of active tension.

The strength of the paper lies in the comparison of results from the DT-MRI derived fibre field, RBM fibre field, and RBM fibre field + dispersed active tension. However, the RBM fibre fields are derived from the fibre angles measured from DT-MRI. Therefore, each fibre field investigated relies on DT-MRI measurements. Therefore, it would have been interesting to perform an additional simulation involving an RBM fibre field, similar to RBM^{uni}, with angles derived from literature, rather than from the DT-MRI field itself. Then, the calculated b_1 and b_2 values could be used to determine n_f , n_s , and n_n dispersion terms for the RBM which is not based on DT-MRI measurements. It would be interesting to demonstrate whether or not similar results could be obtained without any knowledge of the patient-specific fibre field, as is the case when working with clinical cardiac MR data.

Additionally, the rule-based methods only incorporate varying fibre angles. How would results compare to the RBM^{uni} + dispersion in active tension if a sheet angle were also included as other studies have done? It would be more interesting to understand the clear benefits of using a fitted model of dispersion in active tension over including a rule based sheet angle varying over the myocardium.

I think that the paper provides a contribution to the field of personalized cardiac modelling. However, it can be greatly improved upon by incorporating these two additional comparisons discussed above as well as the corrections given below.

General Concerns:

1. It is an interesting point to illustrate the impact of varying n_f , n_s , and n_n (Fig 8). However, each value should be explicitly stated. For example, in Fig 8b, when $n_s = 0.4$, is $n_n = 0.0$?
2. In general, I don't think that having the RV in this study adds anything to the conclusions, particularly since there are no RV fibres measured from DT-MRI and no dispersion in the active tension was added to the RV. It should be made more clear to the reader why a biventricular model was used.
3. Page 6 – The endocardial and epicardial angles used in the RBM^{uni} model should be stated. How different are these angles from those typically reported in literature and used in other rule-based fibre fields?
4. In the RBM¹⁷ model, are the fibre fields smoothly varying between segments? Is there some interpolation between Gauss points? Or is there a discontinuity in the fibre field? Please give reasoning.

5. In Fig 4a, why are there fibre angles around -60 deg on the endocardial surface? Is this due to partial volume effects in endocardial pixels? Are these in a particular region? Could this be due to a fitting error? Has this been investigated? Were these values used when calculating the mean for each region (Table 1)?
6. In the discussion, please clarify the comparison between other values of n_s , n_n and n_f reported in literature. Use the same nomenclature throughout, even if other papers have used different values. Also, it should be noted that the reason why the estimated value of n_s in Sack et al. was so much lower is due to the fact that they used a realistic fibre orientation (including sheet angle variation). Are they really using a different definition of n_s ?

Specific Points:

1. Overall, check grammar everywhere. Numerous mistakes were found throughout.
2. Avoid informal language such as 'Nowadays' (pg. 3, line 16)
3. 'rule based' should be 'rule-based' (pg. 3, line 31)
4. 'out-plane' should be 'out-of-plane' (pg.3, line 52)
5. Correct sentence: pg.3, lines 54-56. It contains numerous grammatical errors.
6. Fig 1b – It appears as though the fibres are nearly exactly circumferential. To illustrate variation more clearly, can the colour indicate the fibre angle? (e.g. blue to red)
7. Fig 2d-f – why do the RV fibre fields look different (primarily the colour labelling)? The RV fibre fields should be precisely the same for each. Are the colour scales exactly the same for each plot? This should be checked.
8. Pg 7, line 48 – 'Bulk Modulus' should be 'bulk modulus'
9. Fig 7 – Confusing wording 'long-axis' versus 'longitudinal axis'. Please clarify.
10. Fig 8a – Colours in the legend do not match the colours in the plot.
11. Add text to introduce Fig 8 and the fact that varying values of n_s and n_n were implemented to test their impact on the pressure volume loop and ejection fraction. Also, please state/explain all values explicitly. For example, in Fig 8b, when $n_s = 0.4$, what are the values of n_n and n_f ?
12. Fig 9d – Change the colour scale to show more variation.
13. Fig 9c (and 6c) – Transmural stress distributions – are these averaged over the entire left ventricle or taken from a mid-ventricle region? If the plot can be updated to include error bars, this would also be helpful information.
14. Page 15, line 44 – Change 'have better pump function' to state 'have a higher ejection fraction'.
15. Page 16, line 17 – ' $n_s = 0.$ ' Should be ' $n_s = 0.0$ '
16. Pg 16 – replace 'RBM-based' with 'rule-based' (two times)

Appendix B

We would like to thank the reviewers for their constructive comments. In the following, we provide point-to-point responses. The original comments are in *italic*. All the corrections in the manuscript are highlighted with blue color.

Referee: 1

This study investigates the differences in ventricular pump function as well as left ventricular end- systolic stresses and strains due to using fibres measured from DT-MRI versus rule-based methods (RBM) in a biventricular model. In particular, the authors illustrate how incorporating a non- symmetrical dispersed active tension model with their RBM fibre field results in pressure-volume loops (for LV and RV) which are similar to those obtained using fibres from DT-MRI. Additionally, the resulting angle of apical tilt is also more similar when incorporating the non-symmetrical dispersion of active tension.

Q1: *The strength of the paper lies in the comparison of results from the DT-MRI derived fibre field, RBM fibre field, and RBM fibre field + dispersed active tension. However, the RBM fibre fields are derived from the fibre angles measured from DT-MRI. Therefore, each fibre field investigated relies on DT- MRI measurements. Therefore, it would have been interesting to perform an additional simulation involving an RBM fibre field, similar to RBMuni, with angles derived from literature, rather than from the DT-MRI field itself. Then, the calculated b1 and b2 values could be used to determine n_f , n_s , and n_n dispersion terms for the RBM which is not based on DT-MRI measurements. It would be interesting to demonstrate whether or not similar results could be obtained without any knowledge of the patient-specific fibre field, as is the case when working with clinical cardiac MR data.*

Response: Thanks for the very constructive suggestion. In the revision, we added extra cases using rule-based fibres with literature-based fibre rotation angles [1] and fibre dispersion parameters from Sommer's study of human ventricular myocardium [2]. Results have been added in Pages 15 - 16, and it reads:

It is interesting to see if similar results could be obtained without any knowledge of the patient-specific fibre field. To this end, we run extra simulations based on RBM^{uni} using literature-based values for n_f , n_s , and n_n . Specifically, we consider (1) no dispersion $n_f = 1$, $n_s = n_n = 0$, (2) $n_f = 0.879$, $n_s = 0.009$, $n_n = 0.112$ [46], and (3) $n_f = 0.646$, $n_s = 0.086$, $n_n = 0.268$, derived from DT-MRI in this study. The fibre rotation angles are also chosen from $30^\circ \sim -30^\circ$ (exRBM¹), $45^\circ \sim -45^\circ$ (exRBM²), or $60^\circ \sim -60^\circ$ (exRBM³) [16]. The results are summarized in Fig. 10 (the main text) in terms of the LV and the RV ejection fractions. Clearly, EFs increase with fibre rotation angles, as more myofibres are aligned longitudinally which enhances the active contraction. Different dispersion parameters also affect the pump function. Compared to case LDDMM, the EFs are lower in exRBM¹ (39.37% (LV), 45.89% (RV)), and still lower in exRBM² (47.86% (LV), 51.72% (RV)). Only exRBM³ with DT-MRI derived dispersion parameters can achieve the similar pump functions as in case LDDMM, though the myofibre rotation angles ($60^\circ \sim -60^\circ$) are much greater than case LDDMM (mean angles $40^\circ \sim -30^\circ$). This would suggest that subject-specific myofibre structure is necessary for cardiac mechanic modelling, as using literature-based myofibre structures seem to underestimate the pump function.

Q2: Additionally, the rule-based methods only incorporate varying fibre angles. How would results compare to the RBMuni + dispersion in active tension if a sheet angle were also included as other studies have done? It would be more interesting to understand the clear benefits of using a fitted model of dispersion in active tension over including a rule based sheet angle varying over the myocardium.

Response: Based on RBM^{uni} case with dispersed active tension, we further test three sets of sheet rotation angles, they are $30^\circ \sim -30^\circ$, $45^\circ \sim -45^\circ$, $60^\circ \sim -60^\circ$ from endocardium to epicardium. We find the sheet rotation angles don't affect ventricular pump function much compared to the fibre rotation angles, which is consistent with other studies [1, 3]. Discussion on sheet rotation angles is added in Discussion section, 3rd paragraph, at page 16. It reads:

Unlike the myofibres which rotate from endocardium to epicardium, here the sheet direction is assumed to align to the radial direction in all cases. In other words, the sheet rotation angle is zero. To evaluate this assumption, we have tested three sets of sheet rotation angles as in [16]: $30^\circ \sim -30^\circ$, $45^\circ \sim -45^\circ$, $60^\circ \sim -60^\circ$, based on case RBM^{uni} with dispersed active tension. The results show that the sheet rotation angle has little effect on ventricular pump function, and the differences in EF between different sheet rotation angles are within 1%. This agrees with observations from other groups. For example, Wang et al [16] found that the sheet rotation angle nearly has no influence on passive mechanics in an LV model.

I think that the paper provides a contribution to the field of personalized cardiac modelling. However, it can be greatly improved upon by incorporating these two additional comparisons discussed above as well as the corrections given below.

Response: Thanks for the supportive comment. These two additional comparisons now have been addressed in this revision, as detailed in our responses above.

Q3: It is an interesting point to illustrate the impact of varying n_f , n_s , and n_n (Fig 8). However, each value should be explicitly stated. For example, in Fig 8b, when $n_s = 0.4$, is $n_n = 0.0$?

Response: Figure 8 has now been updated with n_f , n_s and n_n values. Please also see the first paragraph under Results (b) RBM^{uni} with cross-fibre active tension at page 13.

Q4: In general, I don't think that having the RV in this study adds anything to the conclusions, particularly since there are no RV fibres measured from DT-MRI and no dispersion in the active tension was added to the RV. It should be made more clear to the reader why a biventricular model was used.

Response: The reasons of including the RV in this study are: (1) The RV dynamics will affect the LV dynamics as found in [3] when compared to a stand-alone LV model because of extra loading in the septum and constrains in the LV-RV insertion locations; (2) the LDDMM framework requires geometrical features for warping two different geometries, a bi-ventricular model has much richer information compared to an LV model, which will ensure a more consistent mapping from the canine model to the porcine model. Because

of lack of experimental data from the canine study, we have not varied the RV fibre structures but using a rule-based structure. We now have made it clear in Discussion on the choice of bi-ventricular geometry in this study, see the beginning of page 18. It reads:

Due to lack of DT-MRI data for the RV from the canine experiment, a rule-based approach is used for generating fibre structure in the RV, and zero cross-fibre contraction is assumed. This can be readily improved if the RV fibre structure can be measured as in the LV. We notice there is a difference in the RV systolic function even though the RV model is identical in all three cases. In particular, the RV contracts more in case LDDMM than in the two RBM cases. We think this is due to different LV contraction in the three cases. For instance, the end-systole angle between the long-axis and longitudinal axis is different in each case. Palit et al. [20] also found that there are strong interactions between LV and RV dynamics in diastole. This highlights the importance of LV-RV interaction on cardiac pump function, which is one of the reasons that the RV is included in the bi-ventricle model. In addition, the LDDMM framework [42] relies on geometrical features for warping two different geometries, a bi-ventricular model has much richer information compared to a stand-alone LV model, in particular, those features within the RV-LV insertion regions.

Q5: *Page 6 – The endocardial and epicardial angles used in the RBMuni model should be stated. How different are these angles from those typically reported in literature and used in other rule-based fibre fields?*

Response: The fibre rotation angles are stated when introducing case RBM^{uni} at beginning of page 6.

The values for case RBM^{uni} (endocardium:40°, epicardium:−30°) are within the experimentally reported range [5, 6, 7]. We further summarize a few published myofibre rotation angles from both experimental and numerical studies in Table.A2, the supplementary material. Further discussion can be found in Page 18. It reads:

It is expected that there are differences in myofibre structure between the porcine heart and the canine heart, but this is difficult to assess as we don't have measured DT-MRI fibre structure for the porcine heart. However, despite species difference, we find that the mapped canine myofibre structure agrees well with other studies in terms of mean values [6,24,26], see Table 1. For instance, Ahmad et al [26] measured myofibre rotation angles in LV free wall of neonatal hearts (Anterior $51.1 \pm 3.8^\circ \sim -51.1 \pm 3.8^\circ$, Posterior $40.2 \pm 2.9^\circ \sim -40.2 \pm 2.9^\circ$). Sack et al [6] reported fibre rotation angles for a normal adult porcine heart based on DT-MRI measurements (endocardium: $66.5 \pm 16.6^\circ$, epicardium: $-37.4 \pm 22.4^\circ$). Myofibre rotation angles from published experimental and numerical studies are also summarized in the supplementary material.

Q6: *In the RBM17 model, are the fibre fields smoothly varying between segments? Is there some interpolation between Gauss points? Or is there a discontinuity in the fibre field? Please give reasoning.*

Response: In RBM^{17} case, the fibre fields are not smoothed between different segments. The reasons are (1) the variations of fibre rotation angles among adjacent segments are within the variations of fibre angles from DT-MRI data shown in Figure 4(a) in the main text; (2) the dynamics from case RBM^{17} are very close to case RBM^{uni} . Thus we expect a smoothed fibre field may not show much difference than the unsmoothed one. Moreover, in the latter study, we have focused on the differences between case RBM^{uni} and case LDDMM. Considering the complexity of extra procedures of defining the transition regions between neighbouring segments, we decide not to smooth fibre fields for RBM^{17} case. We now have made it clear when introducing case RBM^{17} on page 6. It reads:

We also have not smoothed rotation angles between segments since those variations are within the range of local angle variations in case LDDMM as suggested in Fig. 4(a)

Q7: In Fig 4a, why are there fibre angles around -60 deg on the endocardial surface? Is this due to partial volume effects in endocardial pixels? Are these in a particular region? Could this be due to a fitting error? Has this been investigated? Were these values used when calculating the mean for each region (Table 1)?

Response: The definition of fibre angles is shown in Fig. 1. After projecting a fibre \mathbf{f}_0 into the $\mathbf{c}_0 - \mathbf{l}_0$ plane,

Figure 1: Definition of in-plane and out-of-plane angle. The positive values means projection of fibre are above circumferential axis and the negative values mean they are below circumferential axis.

we have \mathbf{f}_0^{\parallel} . If \mathbf{f}_0^{\parallel} lies in the first quadrant as shown in Fig. 1, then the rotation angle is positive. If \mathbf{f}_0^{\parallel} in the fourth quadrant ($+\mathbf{c}_0$ and $-\mathbf{l}_0$), then the rotation angle is negative. In a similar way, we define the sheet rotation angle by projecting \mathbf{f}_0 into the $\mathbf{l}_0 - \mathbf{r}_0$ plane, and the rotation angle is defined by the angle between \mathbf{f}_0^{\perp} and $+\mathbf{l}_0$. Large variations of fibre rotation angles appear mainly near the endocardium and epicardium. We have checked the fitting process from the LDDMM framework, and tried with different settings, and the mapped fibre structures are consistent. We are not sure whether those negative angles near endocardium are caused by partial volume effects or the intrinsic noise from the MRI scanner, or if those are the real fibre

structures. We further counted the fibres with negative angles near endocardium across the whole LV, it only accounts of around 1.5% of total fibres. Please refer to Fig.19 in Sommer’s study [2], it can be found that there are fibres with negative angles through the thickness of the cardiac wall. Therefore, we have used all derived fibre angles for calculating fibre rotation angles and dispersion distributions. Clarification is added in the captions of Figure 2 and Figure 4 (the main text).

Q8: *In the discussion, please clarify the comparison between other values of n_s , n_n and n_f reported in literature. Use the same nomenclature throughout, even if other papers have used different values. Also, it should be noted that the reason why the estimated value of n_s in Sack et al. was so much lower is due to the fact that they used a realistic fibre orientation (including sheet angle variation). Are they really using a different definition of n_s ?*

Response: We further compared our values of cross-fibre contraction with other numerical and experimental studies in the discussion. Based on personal communications, Guccione and co-workers have used a different definition of n_s compared to this study, and the same nomenclature is used when comparing with their studies. The lower value of n_n in Sack’s study compared to other and our studies could be because of (1) subject variation; (2) higher $n_f = 1.0$ used in Sack’s study than ours ($n_f = 0.646$), which suggests a higher contraction force along the average myofibre direction. Therefore, to achieve a similar pump function, the cross-fibre contraction could be less since we have found that cross-fibre contraction can facilitate myocardial contraction; (3) n_n and T_a were inversely determined in Sack’s study. As we know that to uniquely determine several parameters from limited data using highly nonlinear models can be very challenging.

They have not mentioned whether they have incorporated sheet angle variation.

Further discussion can be found at page 17. It reads:

We now compare our values of cross-fibre proportions ($n_s = 0.086$, $n_n = 0.268$, $n_f = 0.646$) with previous studies. Based on the experimental study by Lin and Yin [39], Guccione and co-workers introduced cross-fibre active contraction with $n_s = 0.0$, $n_n = 0.4$ and $n_f = 1.0$ [38]. In a recent study, Sack et al [6] inversely determined cross-fibre contraction ratios¹ in a healthy porcine heart ($n_n = 0.07$) and a failure heart ($n_n = 0.14$) with $n_f = 1.0$ and $n_s = 0$. In our study n_n (0.268) is higher than that of Sack’s study [6]. This could be due to (1) subject variation; (2) higher $n_f = 1.0$ used in their study (our $n_f = 0.646$), leading to a higher contraction along the averaged myofibre direction so a lower n_n could match the measured pump function; (3) they inversely determined n_n and T_a , which are not from measurements. In this study, proportions of cross-fibre contraction are derived directly from intrinsic fibre structures, which have a clear biological explanation. When normalized by n_f , the ratio between the sheet-normal and myofibre direction is 41%, which agrees with the ratio reported by Lin and Yin (40%) [39]. We further calculate the dispersion parameters from a recent study on neonatal porcine heart by Ahmad et al [26], $n_f = 0.68$ and $n_n = 0.32$ with nearly negligible $n_s \approx 0.0009$, again very close to our values in this study. We are not aware of any available experimental measurements for estimating n_n and n_s in the myocardium.

¹Note that in their paper they used notation n_s for n_n due to a different definition

Q9 Specific Points:

1. Overall, check grammar everywhere. Numerous mistakes were found throughout.

We have thoroughly proofread the manuscript again by all authors

2. Avoid informal language such as ‘Nowadays’ (pg. 3, line 16)

Corrected

3. ‘rule based’ should be ‘rule-based’ (pg. 3, line 31)

Corrected

4. ‘out-plane’ should be ‘out-of-plane’ (pg.3, line 52)

Corrected

5. Correct sentence: pg.3, lines 54-56. It contains numerous grammatical errors.

Corrected

6. Fig 1b – It appears as though the fibres are nearly exactly circumferential. To illustrate variation more clearly, can the colour indicate the fibre angle? (e.g. blue to red)

The figure now is updated follow the reviewer’s suggestion in page 4.

7. Fig 2d-f – why do the RV fibre fields look different (primarily the colour labelling)? The RV fibre fields should be precisely the same for each. Are the colour scales exactly the same for each plot? This should be checked.

The figure has been updated in page 6. The fibre structures in the RV are same for all cases, but the Pavaview software, used for visualization purpose, will automatically determine the number of fibre tracts represented by tubes, which we can not control. It may appear slightly different but the fibre structures in the RV are identical for all three cases.

8. Pg 7, line 48 – ‘Bulk Modulus’ should be ‘bulk modulus’

Corrected

9. Fig 7 – Confusing wording ‘long-axis’ versus ‘longitudinal axis’. Please clarify.

Definitions of “long-axis” and “longitudinal axis” are added in the caption of Fig 7 at page 14. It reads:

the long-axis (linking the LV basal centre and the LV apex), and the longitudinal axis is represented by the solid line passing the LV basal centre and perpendicular to the basal plane.

10. Fig 8a – Colours in the legend do not match the colours in the plot.

The figure now is updated follow the reviewer’s suggestion in page 14.

11. Add text to introduce Fig 8 and the fact that varying values of n_s and n_n were implemented to test their impact on the pressure volume loop and ejection fraction. Also, please state/explain all values explicitly. For example, in Fig 8b, when $n_s = 0.4$, what are the values of n_n and n_f ?

Values of n_n and n_f are clarified in Figure 8 at page 14. Cases of Figure 8 is introduced at the beginning of the section Result (b), page 13. It reads:

Based on case RBM^{uni}, five different sets of n_s and n_n are chosen to investigate how they affect bi-ventricular dynamics. These are (1) $n_s = 0, n_n = 0$, (2) $n_s = 0.2, n_n = 0$, (3) $n_s = 0.4, n_n = 0$, (4) $n_s = 0.0, n_n = 0.2$, (5) $n_s = 0.0, n_n = 0.4$. For all simulations $n_f = 1.0$. Fig. 8 shows the pump functions with varied n_s or n_n .

12. Fig 9d – Change the colour scale to show more variation.

The figure now is updated follow the reviewer’s suggestion.

13. Fig 9c (and 6c) – Transmural stress distributions – are these averaged over the entire left ventricle or taken from a mid-ventricle region? If the plot can be updated to include error bars, this would also be helpful information.

The transmural stress is averaged across the entire left ventricle, the caption is updated to reflect this. We think adding the error bar to figures 9(c) and 6(c) would be too busy, thus these are unchanged.

14. Page 15, line 44 – Change ‘have better pump function’ to state ‘have a higher ejection fraction’.

Corrected

15. Page 16, line 17 – ‘ $n_s = 0.$ ’ Should be ‘ $n_s = 0.0$ ’

Corrected

16. Pg 16 – replace ‘RBM-based’ with ‘rule-based’ (two times)

Corrected

Referee: 2

The manuscript deals with an important problem related to imaging-based computational models of biological tissues, with particular reference to the cardiac wall and the myocardial fibres distribution.

The paper is well written, and I appreciate it, though I think several weak points need to be addressed before suggesting its publication.

Response: Thanks for the reviewer’s very supportive comment. We now have made major revision according to the reviewer’s suggestions.

Q1: Authors refer to Appendices I do not find in the electronic PDF file. If they are Appendices, why using supplemental material?

Response: The word “appendix” is now replaced with supplementary material.

Q2: The overview of the literature needs to be extended and commented adequately in terms of distributed material model, both passive and active. First, it is well-known that the original fibre distributed model from Gasser et al. suffers several limitations, as shown in many scientific contributions, see e.g. [1-5]. Second, yet in a generalised electro-mechanical framework, several contributions have been proposed accounting both for fibre distribution and multiphysics couplings, see e.g. [6-10]. Third, generalised statistical descriptions of fibre-reinforced biological tissues, accounting for local (in-the-thickness) variation of the microstructures reinforcement have been proposed in the literature, see e.g. [11,12]. A proper description of these studies is in combination with the following comments on the study limitations.

Response: The introduction is expanded extensively to incorporate the work suggested by the reviewer, including the active strain approach, the dispersion work within the active strain approach, and recent development within structural tensor approach. At page 3, it reads:

Later on, Pandolfi and co-workers [30-32] extended Gasser’s general structural tensor approach by including the second order term of the Taylor expansion on the mean invariant along the fibre direction, to improve the accuracy of structural tensor with large dispersions. In a recent study, Melnik [33] further extended the generalised structural tensors to include fibre dispersion in a coupled strain invariant.

Although there are several studies on passive constitutive responses of soft tissue [27,28], very few studies included fibre dispersion in active contraction models for the myocardium. There are two commonly used approaches for modelling active contraction in biological tissue: the active stress formulation [5,6,19,34] and the active strain formulation [9,35,36]. In the active stress formulation, the total stress tensor is decomposed into passive and active parts [37]. This approach has been widely used in personalized cardiac modelling because of its easy implementation, and the fact there are abundant experimental data for the parameter calibration [5,6,19,34]. In the active strain approach, the total deformation gradient \mathbf{F} is multiplicatively decomposed into an elastic part (\mathbf{F}^{pass}) for passive response and an activation part (\mathbf{F}^{act}), which could be more inherent

to the “sliding filament theory” [35]. The same structural tensor for the passive response could be linked to \mathbf{F}^{act} to account for the active response [36]. This seems to be an elegant approach, though fitting personalised parameters to experimental data remains a challenge [35]. It is for this reason that the active stress approach is still adopted here.

Limitations of Gasser’s structural tensor is discussed in section Discussion, etc. In the third paragraph at page 18, it reads:

The spatial variations of the material properties have not been considered in this study, and the same averaged dispersed active contraction model is applied across the whole LV for case RBM^{uni}. This approximation may be reasonable for healthy hearts, but questionable for pathological cases. For example, the myocardium is known to be more heterogeneous post myocardial infarction [57].

Q3: Concerning the adopted active stress electro-mechanical constitutive model, I emphasise once again the scientific contributions already present in the literature, see also [13-17] other than the previous ones. My main concern here is the thermodynamical consistency of the chosen constitutive law. In particular, the orthotropic description assumed should be corroborated by a proper strain energy density to be derived consistently. In the active stress case, this is probably not the case. Accordingly, the Authors must clearly indicate this limitation and refer to generalised thermodynamically consistent approaches.

Response: The active strain approach has been added in the introduction along with active stress approach, including our reasons of choosing active stress formulation in this study. In the second paragraph at page 3.

The polyconvexity issue of the adopted active stress formulation is further discussed in the last paragraph at page 17. It reads:

The convexity of the HO type strain energy function requires all parameters greater than zero as suggested in [63], which is satisfied in our approach. However, as pointed out by Giancesio et al [64], the polyconvexity of the total energy function (passive and active) may not be ensured even though the passive strain energy function is convex. Although we have not experienced stability issues using the active stress approach, we must point out this approach may not be thermodynamically consistent. For generalised thermodynamically consistent approaches, the reader is referred to [9,35,64].

Q4: The assumption of zero cross-fibre contraction for RV is questionable. Why do the authors do not make a parametric analysis for this choice, checking out if their results are robust against it?

Response: Agree. We now added the following simulation and discussion on this point (page 7, last paragraph), it reads:

In this study, we assume the cross-fibre contraction in the RV is zero, i.e. $n_f = 1$, $n_s = 0$, and $n_n = 0$. This is because RV has a much thinner wall thickness, and Ahmad et al. [26] reported the fibre dispersion in the RV is much less than in the LV (9.3° v.s. 19.2°). We also performed simulations for the RBMⁿⁱ case, using the LV’s non-zero cross-fibre contraction for the RV. Our results show the difference of EFs are 0.7% and 4.1% for the LV and RV, respectively. Thus assuming no cross-fibre contraction for the RV seems to be reasonable.

and further mentioned in Discussion, at the beginning of page 18.

Due to lack of DT-MRI data for the RV from the canine experiment, a rule-based approach is used for generating fibre structure in the RV, and zero cross-fibre contraction is assumed. This can be readily improved if the RV fibre structure can be measured as in the LV. We notice there is difference in RV systolic function even though the RV model is identical in all three cases. In particular, the RV contracts more in case LDDMM than in the two RBM cases. We think this is due to the different LV contraction in the three cases. For instance, the end-systole angle between the long-axis and longitudinal axis is different in each case. Palit et al [20] also found that there are strong interactions between the LV and the RV dynamics in diastole. This highlights the importance of LV-RV interaction on cardiac pump function, which is one of the reasons that the RV is included in the bi-ventricle model. Another reason is that the LDDMM framework [42] relies on geometrical features for warping two different geometries, a bi-ventricular model has much richer information compared to a stand-alone LV model, in particular those features within the RV-LV insertion regions.

Q5: In order to provide a more coherent picture of the statistical distribution modelled, I suggest the Authors use three-dimensional plots in-place and out-of-plane as in [12]. The reader would benefit from such a description. Besides, the reproducibility of the constitutive choice would increase. In this case, I suggest the Authors identify the limitations associated with the “average” value of the probability distribution function implied by the Holzapfel et al. approach. It is well-known that for pathological tissues, in which a high value of variance characterises the distribution of the fibres, such an approach is misleading.

Response: A three-dimensional plot is now added in Figure 4 at page 9 following the reviewer’s suggestion. The average assumption is discussed in the third paragraph at page 18, please also refer to Question 2.

Q6: Substantial limitations in terms of chosen boundary conditions need to be discussed in detail, see e.g. [18].

Response: Limitations about boundary conditions have been expanded extensively in the last paragraph of Discussion section at page 18. It reads:

Finally, we would like to mention other limitations of our study. In the boundary conditions we used, the basal plane of the models is constrained along the longitudinal direction, and the rest nodes in the basal plane are free to move. This type of boundary conditions does not represent

in vivo conditions due to lack of the pericardium and great vessels. Under in vivo situation, with the constraints imposed by the pericardium, the apex does not move much. Instead, the basal plane moves downward towards the apex in systole and moves upward in diastole. In a recent study, Pfaller et al. [65] demonstrated that simulated cardiac mechanics could be much closer to the measured heart motion by including the pericardium influences, which highlights the necessary of pericardial-myocardial interaction. A simplified lumped circulation model is used to provide pressure boundary conditions, which is a simplification of pulmonary and systemic circulations. Coupling to a more realistic circulation model, such as one-dimensional systemic models [66,67], will allow us to simulate more detailed cardiovascular function in pathological situations [68]. Furthermore, we have not coupled the blood flow inside ventricle, only applied a spatially homogeneous pressure to the endocardial surface, nor have we considered contraction delay due to the action potential propagation [9]. Tremendous efforts will be needed to address all those limitations, which is beyond the scope of this study.

Q7: *Another major concern to me is the procedure adopted to select material parameters. Due to the high number of parameters to set, it is questionable if such a method returns:*

- 1) *well represented energy (polyconvexity);*
- 2) *robustness against slight variations in the parameters (which is always the case for biological materials);*
- 3) *close-to-uniqueness (stability) of the model.*

Can the Authors provide some convergence analyses in these regards?

Response: 1) The convexity of the HO type strain energy function has been analysed in [25]. Since the passive strain energy function used in this study is in a similar formulation as the original HO model, the convexity can be guaranteed if parameters are all positive for each term. The reason is the convexity of each term additively contributes to the total strain energy function. The initial values for a , b , a_f , b_f , a_n , b_n , a_{fs} , b_{fs} , a_{fn} and b_{fn} are all positive, so the convexity will require the scaling coefficient to be greater than zero. In this study, we used 0.5 for the scaling coefficient. This is now discussed in pages 17, the last paragraph.

2) and 3) The choice of the parameters is further explained in page 17, 4th paragraph. In this study, a , a_f , a_n , a_{fs} , a_{fn} are scaled first to match targeted end-diastolic volumes, respectively. T_{\max} is then determined by achieving targeted ejection fractions. Figure 2 illustrates how the mismatch between the targeted value and the predicted value is reduced by the passive scaling factor and T_{\max} during the inference procedure for the LV. Figure 2(a) shows the mismatch of the LV end-diastolic volume with respect to the scaling factor, which is defined as $|(\text{EDV}^{\text{predict}} - \text{EDV}^{\text{target}})/\text{EDV}^{\text{target}}|$, and figure 2(b) shows the mismatch of the LV ejection fraction ($|(\text{EF}^{\text{predict}} - \text{EF}^{\text{target}})/\text{EF}^{\text{target}}|$) with respect to T_{\max} . From figure 2, it can be seen that both the passive scaling factor and myocardial contractility (T_{\max}) can be nicely determined by matching the targeted values.

Note that this approach will only provide one set of possible parameters, to uniquely infer each parameter of myocardial material property can be extremely challenging due to various difficulties [31], such as limited measured data, and parameter correlation. However, additional sensitivity study of material parameters is now added in the supplementary material.

Figure 2: Relative errors in EDV and EF of the LV when inferring reasonable model parameters by matching targeted end-diastolic volume ($V_0 + 1$) mL with V_0 being the reference volume and ejection fraction 52%.

Q8: *As a final limitation, I suggest the Authors comment extensively on the mismatch between the chosen geometry (porcine) and the chosen fibres dataset (dog). In these regards, it is well-known the close relation between structure and function. Therefore, a clear description of the missing features limiting the generality of the results is strongly required, though a specific software is used for the mapping. Similarly, the angle definition at endocardium and epicardium is referred to the AHA which is a standard adopted for human hearts. All of these different features make the model complex to reproduce. See e.g. [19-21].*

Response: We agree with the reviewer that there will be mismatch between the chosen geometry and the fibre data, which could affect the predicted cardiac function as the reviewer pointed that there is close link between the structure and the pump function. This is a common problem in cardiac modelling community due to limited data available, and usually experimental data from different subjects and species are used for constructing models. In this study, we do not have a full set of DT-MRI data for the porcine heart, thus we have to rely on the freely available canine DT-MRI data. It is very challenging to assess to which extent the used canine DT-MRI derived fibre structure is different from the actual porcine fibre structure, and how it affects the pump function, unless we acquire a full set of experimental data. This remains a future study.

We now added in page 18, 2nd paragraph:

It is expected that there are differences in myofibre structure between the porcine heart and the canine heart, but this is difficult to assess as we don't have measured DT-MRI fibre structure for the porcine heart. However, despite the species difference, we find that the mapped canine myofibre structure agrees well with other studies in terms of mean values [6,24,26], see Table 1. For instance, Ahmad et al [26] measured myofibre rotation angles in the LV free wall of neonatal hearts (Anterior $51.1 \pm 3.8^\circ \sim -51.1 \pm 3.8^\circ$, Posterior $40.2 \pm 2.9^\circ \sim -40.2 \pm 2.9^\circ$). Sack et al [6] reported fibre rotation angles for a normal adult porcine heart based on DT-MRI measurements (endocardium: $66.5 \pm 16.6^\circ$, epicardium: $-37.4 \pm 22.4^\circ$). Myofibre rotation angles from published experimental and numerical studies are also summarized in the supplementary material.

AHA has been widely used for reporting regional cardiac function, especially in human hearts. It is also widely used in animal studies [6] because it provides a standard map among different hearts from different subjects or species. A reference [32] is provided when first introducing AHA segments. In general, all porcine, canine and human hearts have 4 chambers with both left and right hearts. The AHA definition relies on the RV-LV insertion points, and a standard procedure can be followed to determine different segments.

The LDDMM framework [4] is used in this study for warping one geometry to another one, which is publicly available with documentation (<http://www.deformetrica.org>). It uses open-source VTK format for geometry representation. Our own experience suggests that it can be used easily by interested users after some necessary training. We also provide all input files for the LDDMM framework and the Fenics input files for fibre mapping (<https://github.com/HaoGao/FibreGeneration-LDDMM>). We think this will all help readers to reproduce our results. The FEM simulation is done in ABAQUS which is a widely available commercial package. In fact, different FEM packages can replace ABAQUS for the simulations performed in this study, such as Fenics (<https://fenicsproject.org>). All parameter values are provided in the manuscript, therefore, we do not foresee major difficulties for other researchers to reproduce our study.

More discussions to address these points can be found in page 18.

Q9: Finally, I suggest an extensive parametric analysis in which the chosen material parameters lead to specific behaviours.

Response: A sensitivity study is now added with case LDDMM. The so-called “one-point” approach is employed here by varying one parameter at a time and others kept same. Parameters (a , b , a_f , b_f , a_n , b_n , a_{fs} , b_{fs} , a_{fn} , b_{fn} and T_{max}) are first doubled and then halved from the values in Table 3 (the main text). Figure 3(a,b) shows the normalized end-diastolic and end-systolic volumes with respect to case LDDMM. It can be found that the end-diastolic volume is mostly affected by a , b , a_f and a_n , while the end-systolic volume is mostly affected by T_{max} . Figure 3 (c) further shows the changes of ejection fractions. Both the LV and the RV ejection fractions are reduced when doubling a , b , a_f , a_n , and vice versa. T_{max} has the largest effect on the LV and the RV ejection fractions, while other parameters have little influences. Please refer to the supplementary material for the material parameter sensitivity study.

Q10 Minor Concerns:

Notation between small and capital letters for statistical variables has to be appropriately introduced.

Ref 19: missing volume

Ref 24: missing year

Line 10: heart[1,3] space

Response: Done for all.

Q11 Partial List of suggested references:

Response: We have added related references suggested by the reviewer.

Figure 3: Myocardial material parameter sensitivity study, including a , b , a_f , b_f , a_n , b_n , a_{fs} , b_{fs} , a_{fn} , b_{fn} and the active parameter (T_{max}). (a) normalized EDV and ESV values of the LV and (b) the RV with respect to the corresponding baseline values; (c) EF values. The baseline values of the LV and the RV are from the simulation with parameter values in Table 3 (the main text).

References

- [1] Wang H, Gao H, Luo X, Berry C, Griffith B, Ogden R, Wang T. 2013 Structure-based finite strain modelling of the human left ventricle in diastole. *International journal for numerical methods in biomedical engineering* **29**, 83–103.
- [2] Sommer G, Schriefl AJ, Andrä M, Sacherer M, Viertler C, Wolinski H, Holzapfel GA. 2015 Biomechanical properties and microstructure of human ventricular myocardium. *Acta biomaterialia* **24**, 172–192.
- [3] Palit A, Bhudia SK, Arvanitis TN, Turley GA, Williams MA. 2015 Computational modelling of left-ventricular diastolic mechanics: Effect of fibre orientation and right-ventricle topology. *Journal of biomechanics* **48**, 604–612.

- [4] Durrleman S, Prastawa M, Charon N, Korenberg JR, Joshi S, Gerig G, Trouvé A. 2014 Morphometry of anatomical shape complexes with dense deformations and sparse parameters. *NeuroImage* **101**, 35–49.
- [5] Ahmad F, Soe S, White N, Johnston R, Khan I, Liao J, Jones M, Prabhu R, Maconochie I, Theobald P. 2018 Region-Specific Microstructure in the Neonatal Ventricles of a Porcine Model. *Annals of biomedical engineering* **46**, 2162–2176.
- [6] Sack KL, Aliotta E, Ennis DB, Choy JS, Kassab GS, Guccione JM, Franz T. 2018 Construction and validation of subject-specific biventricular finite-element models of healthy and failing swine hearts from high-resolution DT-MRI. *Frontiers in physiology* **9**, 539.
- [7] Helm PA, Tseng HJ, Younes L, McVeigh ER, Winslow RL. 2005 Ex vivo 3D diffusion tensor imaging and quantification of cardiac laminar structure. *Magnetic resonance in medicine* **54**, 850–859.
- [8] Streeter Jr DD, Spotnitz HM, Patel DP, ROSS Jr J, Sonnenblick EH. 1969 Fiber orientation in the canine left ventricle during diastole and systole. *Circulation research* **24**, 339–347.
- [9] Lin D, Yin F. 1998 A multiaxial constitutive law for mammalian left ventricular myocardium in steady-state barium contracture or tetanus. *Journal of biomechanical engineering* **120**, 504–517.
- [10] Wenk JF, Klepach D, Lee LC, Zhang Z, Ge L, Tseng EE, Martin A, Kozerke S, Gorman JH, Gorman RC, Guccione JM. 2012 First Evidence of Depressed Contractility in the Border Zone of a Human Myocardial Infarction. *The Annals of Thoracic Surgery* **93**, 1188–1193.
- [11] Pandolfi A, Vasta M. 2012 Fiber distributed hyperelastic modeling of biological tissues. *Mechanics of Materials* **44**, 151–162.
- [12] Vasta M, Gizzi A, Pandolfi A. 2014 On three- and two-dimensional fiber distributed models of biological tissues. *Probabilistic Engineering Mechanics* **37**, 170–179.
- [13] Gizzi A, Pandolfi A, Vasta M. 2018 A generalized statistical approach for modeling fiber-reinforced materials. *Journal of Engineering Mathematics* **109**, 211–226.
- [14] Melnik AV, Luo X, Ogden RW. 2018 A generalised structure tensor model for the mixed invariant I8. *International Journal of Non-Linear Mechanics* **107**, 137–148.
- [15] Pandolfi A, Gizzi A, Vasta M. 2016 Coupled electro-mechanical models of fiber-distributed active tissues. *Journal of biomechanics* **49**, 2436–2444.
- [16] Gasser TC, Ogden RW, Holzapfel GA. 2005 Hyperelastic modelling of arterial layers with distributed collagen fibre orientations. *Journal of the royal society interface* **3**, 15–35.
- [17] Holzapfel GA, Niestrawska JA, Ogden RW, Reinisch AJ, Schriefl AJ. 2015 Modelling non-symmetric collagen fibre dispersion in arterial walls. *Journal of the Royal Society Interface* **12**, 20150188.
- [18] Gao H, Aderhold A, Mangion K, Luo X, Husmeier D, Berry C. 2017 Changes and classification in myocardial contractile function in the left ventricle following acute myocardial infarction. *Journal of The Royal Society Interface* **14**, 20170203.
- [19] Genet M, Lee LC, Nguyen R, Haraldsson H, Acevedo-Bolton G, Zhang Z, Ge L, Ordovas K, Kozerke S, Guccione JM. 2014 Distribution of normal human left ventricular myofiber stress at end diastole and end systole: a target for in silico design of heart failure treatments. *Journal of applied physiology* **117**, 142–152.

- [20] Eriksson TS, Prassl AJ, Plank G, Holzapfel GA. 2013 Modeling the dispersion in electromechanically coupled myocardium. *International journal for numerical methods in biomedical engineering* **29**, 1267–1284.
- [21] Rossi S, Lassila T, Ruiz-Baier R, Sequeira A, Quarteroni A. 2014 Thermodynamically consistent orthotropic activation model capturing ventricular systolic wall thickening in cardiac electromechanics. *European Journal of Mechanics-A/Solids* **48**, 129–142.
- [22] Quarteroni A, Lassila T, Rossi S, Ruiz-Baier R. 2017 Integrated Heart—Coupling multiscale and multiphysics models for the simulation of the cardiac function. *Computer Methods in Applied Mechanics and Engineering* **314**, 345–407.
- [23] Guccione J, McCulloch A. 1993 Mechanics of active contraction in cardiac muscle: part I—constitutive relations for fiber stress that describe deactivation. *Journal of biomechanical engineering* **115**, 72–81.
- [24] Kung GL, Vaseghi M, Gahm JK, Shevtsov J, Garfinkel A, Shivkumar K, Ennis DB. 2018 Microstructural infarct border zone remodeling in the post-infarct swine heart measured by diffusion tensor MRI. *Frontiers in physiology* **9**, 826.
- [25] Holzapfel GA, Ogden RW. 2009 Constitutive modelling of passive myocardium: a structurally based framework for material characterization. *Philosophical Transactions of the Royal Society A: Mathematical, Physical and Engineering Sciences* **367**, 3445–3475.
- [26] Giamtesio G, Musesti A, Riccobelli D. 2019 A comparison between active strain and active stress in transversely isotropic hyperelastic materials. *Journal of Elasticity* pp. 1–20.
- [27] Pfaller MR, Hörmann JM, Weigl M, Nagler A, Chabiniok R, Bertoglio C, Wall WA. 2019 The importance of the pericardium for cardiac biomechanics: From physiology to computational modeling. *Biomechanics and modeling in mechanobiology* **18**, 503–529.
- [28] Olufsen MS, Peskin CS, Kim WY, Pedersen EM, Nadim A, Larsen J. 2000 Numerical simulation and experimental validation of blood flow in arteries with structured-tree outflow conditions. *Annals of biomedical engineering* **28**, 1281–1299.
- [29] Guan D, Liang F, Gremaud PA. 2016 Comparison of the Windkessel model and structured-tree model applied to prescribe outflow boundary conditions for a one-dimensional arterial tree model. *Journal of biomechanics* **49**, 1583–1592.
- [30] Chen WW, Gao H, Luo XY, Hill NA. 2016 Study of cardiovascular function using a coupled left ventricle and systemic circulation model. *Journal of Biomechanics* **49**, 2445–2454.
- [31] Gao H, Li W, Cai L, Berry C, Luo X. 2015 Parameter estimation in a Holzapfel–Ogden law for healthy myocardium. *Journal of engineering mathematics* **95**, 231–248.
- [32] Cerqueira MD, Weissman NJ, Dilsizian V, Jacobs AK, Kaul S, Laskey WK, Pennell DJ, Rumberger JA, Ryan T et al. 2002 Standardized myocardial segmentation and nomenclature for tomographic imaging of the heart: a statement for healthcare professionals from the Cardiac Imaging Committee of the Council on Clinical Cardiology of the American Heart Association. *Circulation* **105**, 539–542.